# Tig1 regulates proximo-distal identity during salamander limb regeneration

Catarina R. Oliveira[1], Dunja Knapp [1✉], Ahmed Elewa[2], Tobias Gerber[3], Sandra G. Gonzalez Malagon [4,5], Phillip B. Gates[4], Hannah E. Walters [1], Andreas Petzold[1], Hernan Arce[6,7], Rodrigo C. Cordoba [6], Elaiyaraja Subramanian[2], Osvaldo Chara[6,7,8], Elly M. Tanaka [9], András Simon[2] & Maximina H. Yun [1,10✉]

Salamander limb regeneration is an accurate process which gives rise exclusively to the missing structures, irrespective of the amputation level. This suggests that cells in the stump have an awareness of their spatial location, a property termed positional identity. Little is known about how positional identity is encoded, in salamanders or other biological systems. Through single-cell RNAseq analysis, we identified Tig1/Rarres1 as a potential determinant of proximal identity. Tig1 encodes a conserved cell surface molecule, is regulated by retinoic acid and exhibits a graded expression along the proximo-distal axis of the limb. Its over-expression leads to regeneration defects in the distal elements and elicits proximal dis-placement of blastema cells, while its neutralisation blocks proximo-distal cell surface interactions. Critically, Tig1 reprogrammes distal cells to a proximal identity, upregulating Prod1 and inhibiting Hoxa13 and distal transcriptional networks. Thus, Tig1 is a central cell surface determinant of proximal identity in the salamander limb.

[1] Technische Universität Dresden, CRTD/Center for Regenerative Therapies Dresden, Dresden, Germany. [2] Department of Cell and Molecular Biology, Karolinska Institute, Stockholm, Sweden. [3] European Molecular Biology Laboratory (EMBL), Heidelberg, Germany. [4] Institute of Structural and Molecular Biology, University College London, London, UK. [5] Biomedical Research Institute, Foundation for Research and Technology, University of Ioannina Campus, 45115, Ioannina, Greece. [6] Systems Biology Group, Institute of Physics of Liquids and Biological Systems, National Scientific and Technical Research Council (CONICET) and University of La Plata, La Plata, Argentina. [7] Instituto de Tecnología, Universidad Argentina de la Empresa (UADE), Buenos Aires, Argentina. [8] Technische Universität Dresden, Center for Information Services and High Performance Computing ZIH, Dresden, Germany. [9] Institute of Molecular Pathology, Vienna Biocenter, Vienna, Austria. [10] Max Planck Institute for Molecular Cell Biology and Genetics, Dresden, Germany. ✉email: dunja.knapp@tu-dresden.de; maximina.yun@tu-dresden.de

The mechanisms by which cells determine their position within a biological structure are central to the correct development and regeneration of tissues and organs. Despite their importance, they remain poorly understood. Significant insights into this problem originate from the study of salamander limb regeneration, a powerful model for probing the mechanisms underlying positional information[1–3]. In salamanders, the amputation of a limb elicits the generation of a mass of progenitor cells—named blastema—which subsequently grows and re-patterns, invariably giving rise to the structures distal to its level of origin[4]. This indicates that mature cells within the limb stump retain a memory of their location along the proximo-distal (PD) axis which blastema cells are able to interpret in order to regenerate only the missing parts, a property termed positional identity[5]. Elegant blastema juxtaposition experiments offered first clues into the nature of PD identity, suggesting that it is encoded as a proximo-distal gradient of molecules along the limb and manifested at the cell surface level. First, grafting of a distal blastema onto the site of a proximal one leads to the displacement of the distal blastema to its level of origin along the PD axis as regeneration proceeds[6], indicating an initial recognition of PD disparity at the cell surface. Second, apposing proximal and distal blastemas in culture results in the engulfment of the distal blastema by the proximal one, indicating the existence of differential adhesive properties between cells of different PD identities and therefore underscoring the importance of cell surface interactions[7]. Third, grafting of a wrist blastema onto a shoulder stump elicits regenerative growth from the shoulder stump, while wrist blastema cells contribute to the hand structures[8]. Similarly, transplantation of distal blastema cells to proximal blastemas results in translocation of distal cells and their contribution to hand elements, indicating that PD identity is manifested on an individual cell basis and is a stable feature of cell identity[9]. Additional clues into the nature of PD identity came from classical experiments addressing the effect of retinoic acid (RA) and its derivatives on limb regeneration. Notably, exposing a wrist blastema to increasing doses of retinoic acid leads to regeneration of more proximal elements in a dose-dependent manner, resulting in tandem limb duplications at the highest retinoic acid concentrations[5,10]. Thus, retinoic acid is able to reprogramme positional identity towards proximal values.

From these seminal findings, an approach for identifying a mediator of proximal identity was proposed[11]: it would exhibit a graded expression along the PD axis, be present at the cell surface, and be upregulated by retinoic acid. This led to the identification of the GPI-anchored cell-surface molecule Prod1[11] in newts, which fulfils all said criteria. Indeed, newt Prod1 was shown to mediate the engulfment activity of proximal blastemas when juxtaposed to distal ones and thus cell affinity, an important aspect of proximal identity[11]. Subsequently, axolotl Prod1 overexpression in early blastemas was shown to affect the patterning of distal elements during limb regeneration and, notably, its focal overexpression in cells of distal domains led to their translocation to proximal positions within the limb[9]. Further, axolotl Prod1 expression was shown to be regulated by Meis homeobox transcription factors[12], critical targets of retinoic acid during proximo-distal patterning of the vertebrate limb[13]. Altogether, these experiments associated Prod1 to the acquisition of proximal identity. Given its PD distribution along the mature limb, it was proposed that Prod1 is part of the positional memory system that confers mature limb cells their spatial identity and determines the extent of regeneration[14]. As such a system would determine how much to regenerate based on the levels of PD factors, direct or indirect connections to growth control mechanisms are to be expected. Indeed, Prod1 was found to interact with a Schwann-cell-derived factor of the anterograddient protein family, nAG, which acts as a mitogen for blastema cells[15],

providing a link between positional determination and growth regulation.

To date, Prod1 is considered the best example of a molecule encoding positional information[3]. However, important questions remain unanswered. First, it is unclear whether Prod1 reprogrammes cells to a proximal identity, as it could be expected from a PD determinant, or simply regulates their adhesive properties such that they translocate to proximal locations. Second, axolotl Prod1 lacks a membrane anchor, despite exhibiting the ability to regulate aspects of cell affinity associated to proximalisation[16]. This raises the possibility that, in order to achieve a graded distribution along the PD axis and execute its functions, Prod1 interacts with additional, as yet unidentified proximo-distal identity factors. In view of these considerations, we set out to identify components of the PD determination system.

In this work, we identify Tig1 as a PD regulator. We demonstrate that *Tig1* is a RA-responsive gene encoding a cell surface molecule, which exhibits graded expression along the PD axis of the mature limb. We show that Tig1 fulfils functional criteria attributed to a proximal determinant, including the ability to reprogramme blastema cells to a proximal identity. Thus, our results implicate Tig1 in the regulation of proximo-distal identity during salamander limb regeneration.

## Results

**Tig1, a potential cell surface determinant of proximal identity.** To identify molecules involved in the PD identity system, we performed principal component analysis on a single-cell RNAseq dataset of connective tissue cells during axolotl limb regeneration[17]. This particular tissue type was selected given that it is implicated in the establishment of PD identity during limb regeneration, in contrast to other types such as muscle, which appears to be positionally naïve[18]. Through this approach, we identified a principal component (PC3) that modelled the proximo-distal axis based on the expression profile of the key developmental markers Hoxa13+ (distal) and Meis2+ (proximal) cells (Supplementary Fig. 1a, b and Supplementary Data 1, 2). By mining the transcriptome of single cells at the Meis2+ end of this component, we identified the axolotl homologue of *Tig1* (Tazarotene-induced gene 1), also known as *Rarres1* (Retinoic Acid Receptor Responder 1), as a highly enriched gene in modelled proximal cells (Supplementary Fig. 1a–c). A limitation of this dataset is the absence of important PD markers such as Prod1, as it was not originally included in the transcriptome used. To overcome this, we analysed a recently published scRNAseq dataset of axolotl limb regeneration[19], in which we were able to detect Prod1 expression in the connective tissue compartment. As above, we identified a principal component (PC1), which modelled the PD axis based on the distribution of Hoxa13/Hoxd13 (distal) and Meis1/Prod1 (proximal) in connective tissue cells of the early blastema stage (Fig. 1a, b). Further, a uniform manifold approximation and projection (UMAP) performed to visualise cellular heterogeneity revealed a clear separation of cells based on their expression of PD markers (Fig. 1c), with Tig1+ cells found at the same area as those expressing Meis1, Meis2 and Prod1 (Fig. 1c). Next, we calculated 'distal' and 'proximal' scores for each cell based on the expression of Hoxa13/Hoxd13 for the former and Meis1/Meis2/Prod1 for the latter (Fig. 1d and Supplementary Data 3). Notably, Tig1 is differentially expressed along the proximo-distal axis with enrichment in proximal cells (Fig. 1e), raising the possibility that it could constitute a proximal determinant.

To experimentally test this notion, we performed qRT-PCR gene expression studies in mature and regenerating limbs. As suggested by our in silico analysis, Tig1 is expressed in a graded fashion along the PD axis of the mature axolotl limb, reaching the

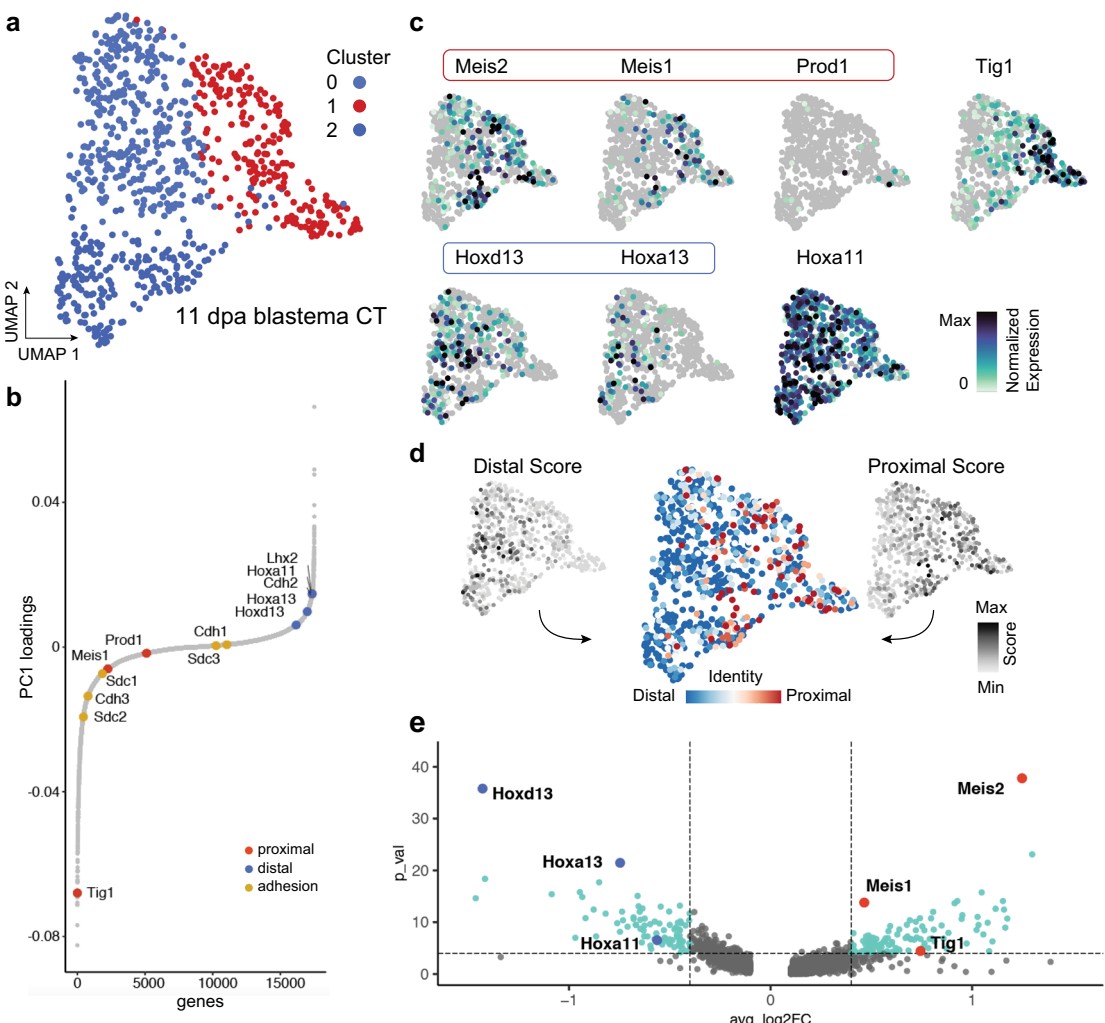

**Fig. 1 Single-cell transcriptomics identifies Tig1 as a marker for proximal limb identity in connective tissue cells of early limb blastemas. a** UMAP embedding of single-cell transcriptomic data from connective tissue cells at 11 days postamputation[19]. This corresponds to an early blastema regeneration stage (8–9 cm snout-to-tail axolotls). Each dot represents a single cell; colours indicate results of Louvain clustering of single cells. **b** Contribution of axolotl genes to PC1. Genes are presented as dots and are ordered based on their contribution to PC1. Genes associated with the proximal (red) and distal (blue) axes, or with cell adhesion (yellow) are indicated. **c** Gene expression for genes of the proximal-distal axis are shown as feature on top of the UMAP embedding. Encircled genes are used to calculate a proximal and distal score (see **d**). **d** Distal (left) and proximal (right) scores were calculated for each cell and visualised on the UMAP embedding. Scores were combined to obtain proximal-distal identity assignments (centre) for each cell as indicated on the embedding. **e** Volcano plot visualisation of genes that are differentially expressed between the 100 most proximal and the 100 most distal cells identified by the scoring in **d**. Relevant proximal (red) and distal (blue) genes are highlighted. $P$ values (Wilcoxon rank-sum test) are shown as $-\text{Log}_{10}$ and the $p$ value cutoff is 0.0001 (horizontal dotted line) and the fold change (FC) cutoffs are 0.4 and −0.4 (vertical dotted lines).

highest level in the upper arm and decreasing towards the hand (Fig. 2a). Of note, this is also observed in the limbs of *Notophthalmus viridescens* newts (Supplementary Fig. 2). Thus, similar to Prod1, Tig1 fulfils a key criterion associated with a molecular determinant of proximal identity, namely graded expression along the mature limb.

Human TIG1 is an RA-responsive putative tumour suppressor[20,21] which exhibits structural resemblance to CD38 (Supplementary Fig. 3a), a retinoid-responsive surface molecule found in immune cells[22]. Comparative protein sequence analysis (Supplementary Fig. 3b) indicates that axolotl TIG1 exhibits substantial sequence similarity (57%) with its human counterpart. Further, the most significant protein features of human TIG1, namely a transmembrane 'latexin' domain -present in the carboxypeptidase inhibitor Latexin and highly conserved among vertebrates-, and a hyaluronic acid-binding motif[21,23], are all present in axolotl TIG1 (Supplementary Fig. 3b). These

observations raise the possibility that axolotl Tig1 is an RA-responsive gene encoding a cell surface protein, thus fulfilling important criteria[11] for a factor involved in the determination of proximo-distal identity. Indeed, axolotl Tig1 expression is upregulated by RA in vivo, in both normal and regenerating limb tissues (Fig. 2b). In agreement, several RA regulatory elements (RXRA; RXRB, RXRG; RARA and RARG) and MEIS binding sites (MEIS1 and MEIS2) are present in the axolotl Tig1 promoter (Supplementary Data 4). Of note, RA-dependent Tig1 induction is observed in connective tissue-derived *Am*AL1 cells, but not in the myogenic line *Nv*A1 (Supplementary Fig. 4). To determine its cellular localisation, axolotl Tig1 was expressed in *Am*AL1 cells as a fusion protein carrying a myc tag in either its C- or N-terminus (Supplementary Fig. 5a), followed by live-cell staining with anti-myc antibodies (Fig. 2c). While no myc-derived signal is detected upon expression of N-terminal myc-Tig1, a distinct cell surface staining is observed following expression of

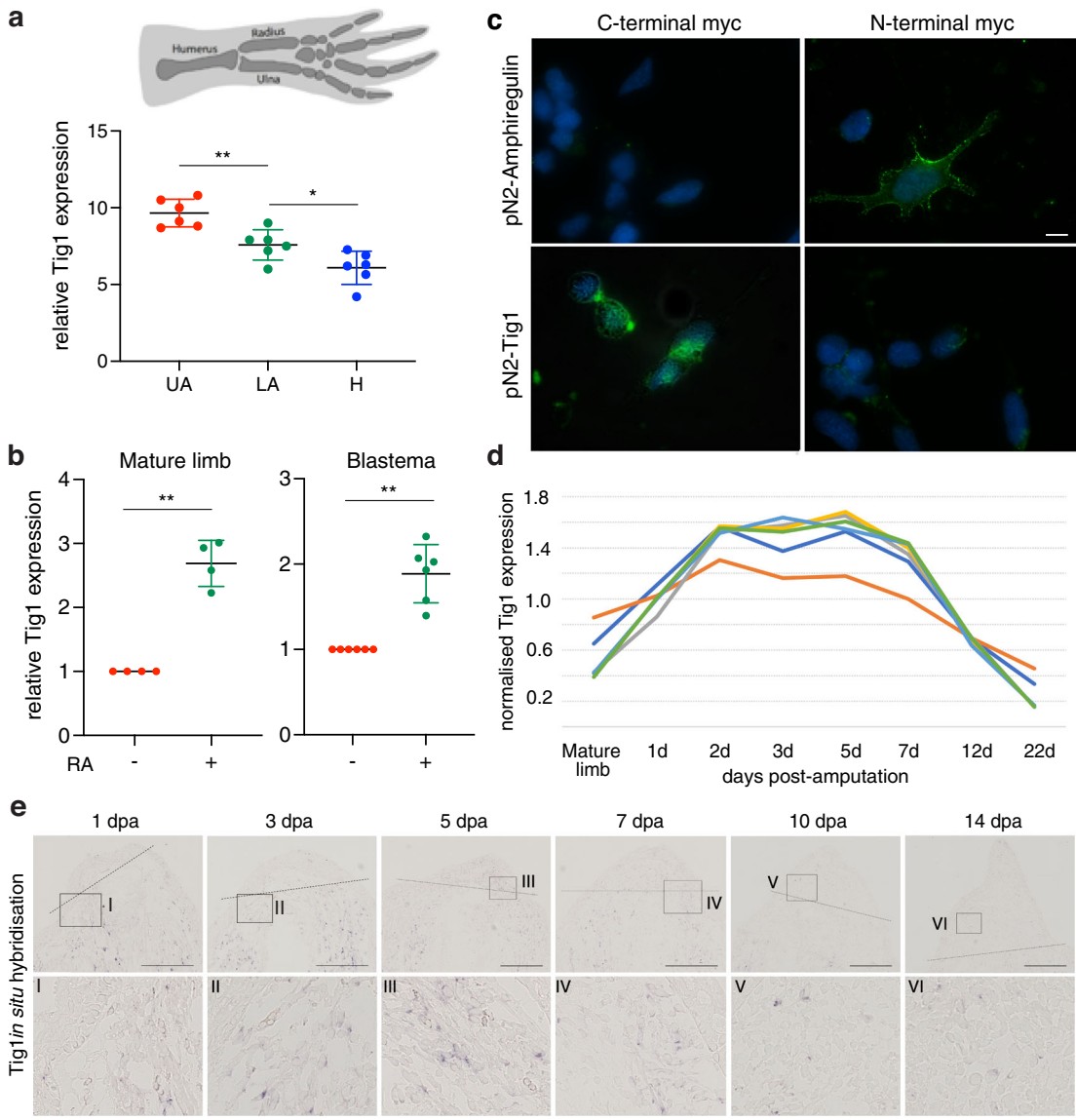

**Fig. 2 The retinoic acid-responsive gene Tig1 encodes a cell surface protein and is differentially expressed along the proximo-distal axis of the axolotl limb. a** Tig1 expression along the proximo-distal axis of the mature axolotl limb, relative to Ef1-α (UA upper arm, LA lower arm, H hand, qRT-PCR, $n = 6$ biologically independent samples). Adj. $p$ values: $^*p = 0.0491$, $^{**}p = 0.0068$ (one-way ANOVA followed by Tukey's multiple comparison test). **b** Expression of axolotl Tig1 in mature or regenerating (mid bud stage) limbs 48 h after retinoic acid (RA) treatment (100 μg/g body weight), relative to DMSO treated samples, normalised against Ef1-α expression; qRT-PCR, $n = 4$ (mature limb) or $n = 6$ (regenerating limb) biologically independent samples. $^{**}p = 0.0025$ (mature limb), $^{**}p = 0.0014$ (blastema) (two-tailed paired $t$-test). **c** Representative images of AL1 cells after electroporation with the indicated myc-tagged constructs and live-cell staining using anti-myc antibodies. Scale bar: 20 μm. **d** Microarray[54] analysis of Tig1 gene expression for the indicated bulk tissues and days post limb amputation. Note the upregulation of Tig1 during the first 7 days after amputation. Results were obtained with six independent microarray probes, each one showing the mean of three biological replicates. Note the consistent expression profile obtained with each one. **e** Expression of axolotl Tig1 during limb regeneration from an upper arm amputation (7.5 cm snout-to-tail axolotls) by in situ hybridisation. Representative images show blastema (upper row) or corresponding magnified area (lower row). No signal is detected in the mature limb using ISH. Scale bar: 500 μm. Each experiment was independently repeated three times with similar results (**c**, **e**). Error bars represent the standard error of the mean (SEM) (**a**, **b**). Source data are provided as a Source Data file.

the C-terminal tagged Tig1 (Fig. 2c). The opposite pattern is observed for myc-tagged forms of axolotl Amphiregulin, a known surface protein whose N-terminus faces the extracellular space (Fig. 2c). Similar results were observed in 293 T cells (Supplementary Fig. 5b). Together, these observations indicate that axolotl TIG1 localises to the cell surface, with its C-terminus, which comprises the majority of its protein sequence, facing the extracellular space.

While little is known about the biological functions of Tig1 in any context, in vitro studies suggest that human Tig1 has a

negative impact on cell proliferation[21]. This is particularly interesting in light of the fact that Prod1 has also been shown to impair proliferation[9]. To test whether this is also the case for axolotl Tig1, pN2-Tig1 or empty pN2 vectors were electroporated together with a fluorescent reporter into mid bud blastemas, followed by systemic EdU injection and subsequent analysis of EdU incorporation (Supplementary Fig. 6a). As the latexin domain had been proposed to be important for Tig1 function[23], we generated a Tig1 mutant version (Tig1^P155A) in which the critical cis-proline residue within the latexin domain was mutated

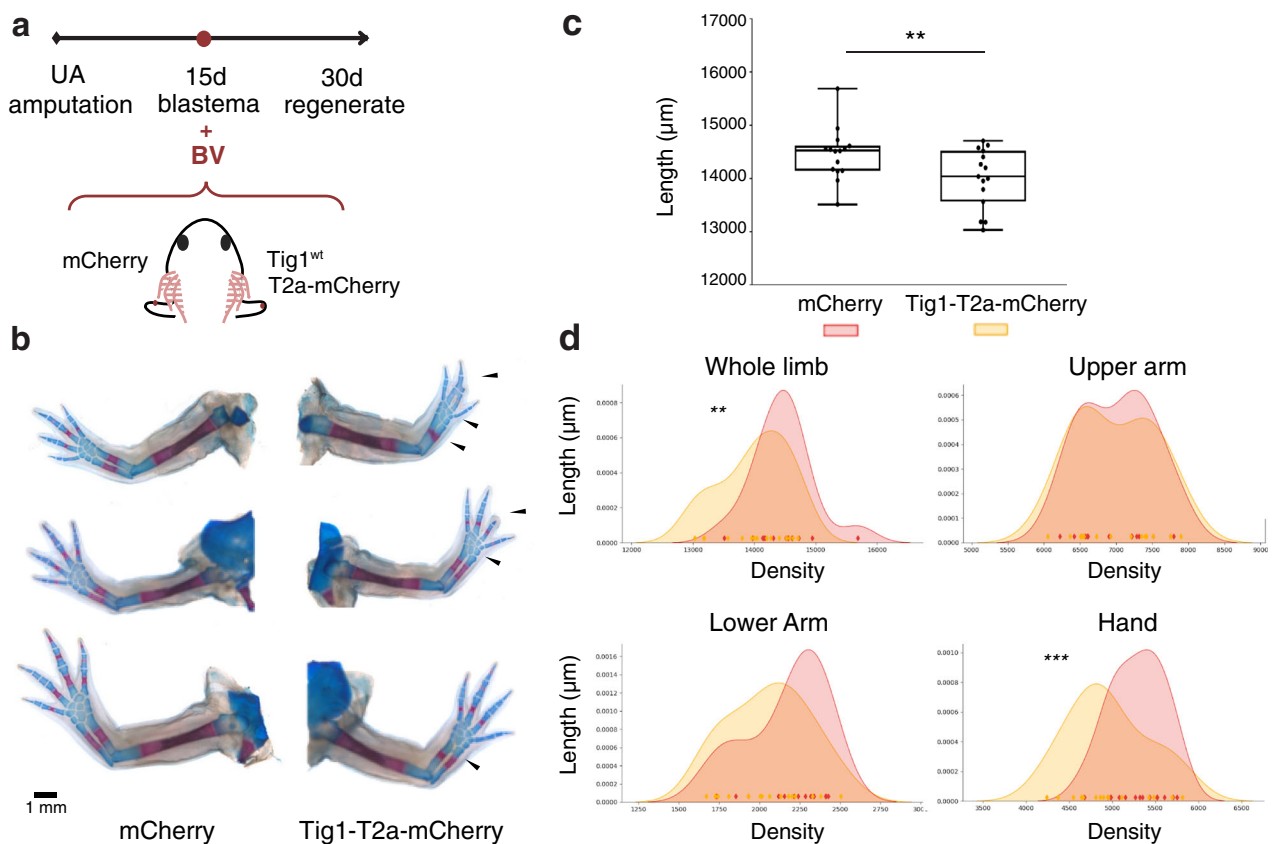

**Fig. 3 Tig1 overexpression disrupts regeneration of distal elements. a** Schematic of the experimental approach. **b** Tig1 Bv transduction from mid bud blastema onwards leads to shortening of the regenerated limb through malformations in the distal limb elements, as analysed through Alcian blue/Alizarin red staining ($n = 15$ biologically independent animals). **c** Quantification of limb length. The box horizontal boundaries define the interquartile range (upper boundary: 75th percentile and bottom boundary: 25th percentile of dataset values), the internal line indicates Median, whiskers indicate highest and lowest value for each condition. **$p = 0.0014$ (two-tailed paired-matched $t$-test). **d** Kernel density estimation of lengths for the indicated limb segment. The coloured diamonds on the x axis represent individual measurements. **$p = 0.0014$; ***$p = 0.0004$ (two-tailed paired-matched $t$-test). A broadening of the distribution towards shorter lengths is observed upon Tig1 overexpression for whole limb, lower arm and hand, while no changes are observed in the upper arm. Source data are provided as a Source Data file.

to an alanine. Overexpression of either Tig1 or Tig1[P155A] led to significant decreases in EdU incorporation by blastema cells (Supplementary Fig. 6b, c), suggesting that the function of Tig1 as a negative regulator of cell proliferation is conserved in the axolotl and that an intact latexin domain is dispensable for it. Together, the aforementioned observations draw important parallels between Tig1 and Prod1, and thus pinpoint Tig1 as a candidate regulator of proximal identity during regeneration.

**Tig1 misexpression disrupts regeneration of distal structures.** To gain insights into the functions of Tig1, we first examined its expression throughout limb regeneration. Notably, Tig1 expression is upregulated during the first 7 days postamputation (dpa, Fig. 2d, e and Supplementary Fig. 7), coinciding with the establishment of proximal identities[24], followed by a strong downregulation that is maintained until the end of the process (Supplementary Fig. 8), in agreement with the previous reports[17]. These observations are recapitulated by in situ hybridisation analysis, which also reveals that Tig1 expression is restricted to the proximal blastema and adjacent stump tissue (Fig. 2e and Supplementary Fig. 9). Regeneration of lower and upper arms follow a similar pattern, however, Tig1 levels start to decrease earlier (noticeable at 10 dpa) in the lower arm (Supplementary Figs. 10, 11). Further, Tig1 expression is primarily detected in connective tissue cells (Supplementary Fig. 12), as suggested by our transcriptomic analysis (Fig. 1 and Supplementary Fig. 1).

Next, we sought to disrupt the endogenous expression dynamics by overexpressing Tig1 from the mid bud stage onwards, when it is normally downregulated (Fig. 3a). Significantly, increasing the levels of Tig1 via baculovirus(Bv)-mediated transduction, which preferentially targets connective tissue cells[25], leads to impaired regeneration of distal elements (Fig. 3a, b). Defects induced by Tig1 overexpression include fusion and shortening of digits (5/21), morphological alterations in the lower arm elements radius and ulna, namely the uneven thickness of the elements (2/21) and elongated or fused distal epiphysis (6/21), as well as malformations of carpals and metacarpals (11/21), while normal limb elements are observed in contralateral control limbs transduced with a mCherry Bv. Importantly, Tig1 transduction leads to a significant reduction in limb length (Fig. 3c), which is accompanied by shortening of the lower arm and a significant downsizing of the hand, while the length of the upper arm does not differ from the control condition (Fig. 3d). This phenotype suggests a specific effect of Tig1 in the regeneration of distal elements. Furthermore, it exhibits a striking resemblance to that resulting from overexpressing Prod1[9] during axolotl regeneration. Taken together, these observations raise the possibility that Tig1 may be implicated in the determination of proximal identity.

**Tig1 induces proximal displacement of blastema cells.** The few molecules implicated in the regulation of proximal identity to date, Prod1 and Meis1/2, are able to elicit the proximal

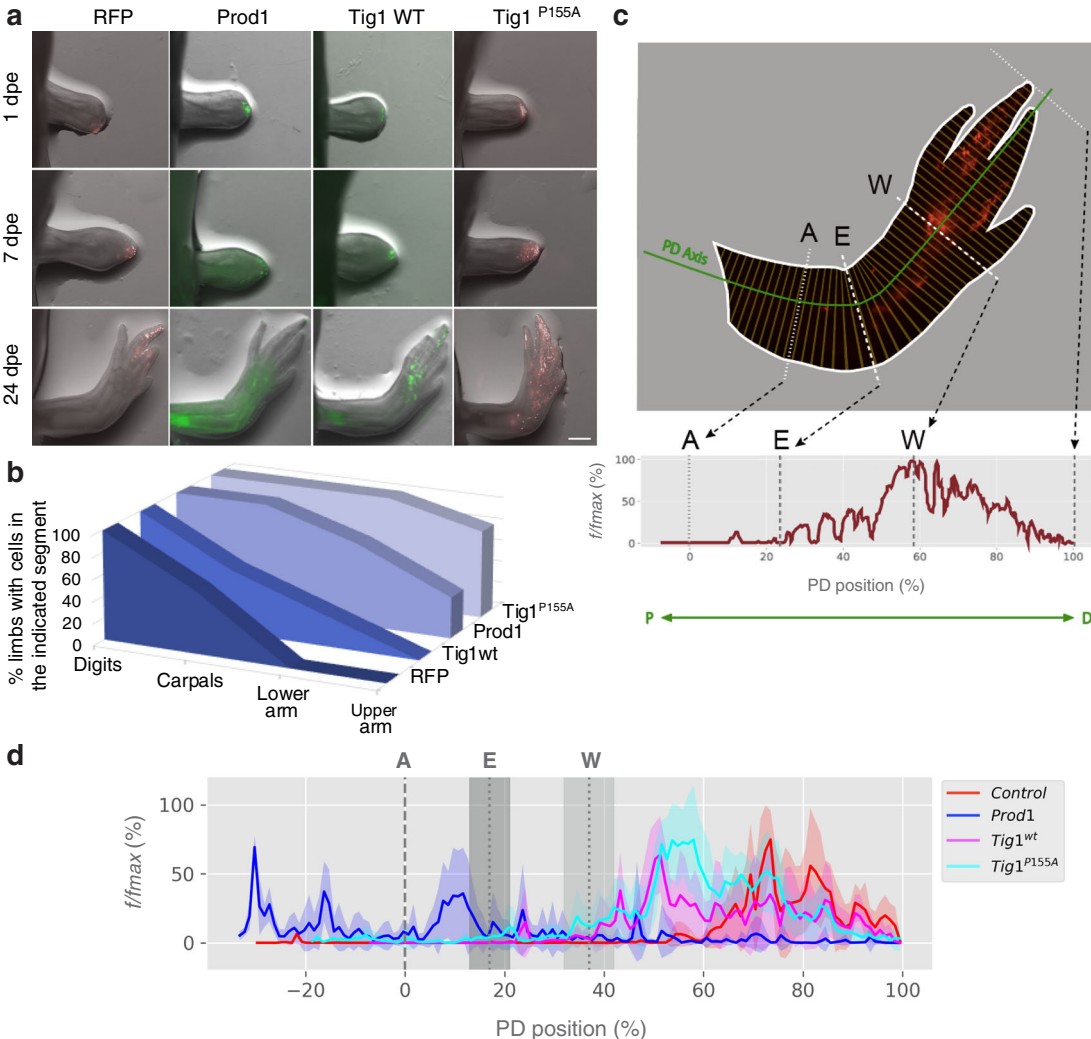

**Fig. 4 Tig1 induces distal-to-proximal displacement of blastema cells. a** Representative images of the displacement assay. Cells in the distal third of a 4-day blastema give rise to distal elements[9]. Upon electroporation of a control vector ($n = 7$) to enable tracing during regeneration, they contribute primarily to digits (first column). Overexpression of Prod1 ($n = 8$), Tig1 or Tig1$^{P155A}$ ($n = 13$) coupled to fluorescent proteins promotes autologous displacement of cells from distal to proximal positions during limb regeneration. dpe days post electroporation. Scale bar: 1 mm. **b** Quantification of the percentage of animals with electroporated cells found in the indicated limb segments. **c** An image analysis algorithm for the automatised quantification of cells (by fluorescence intensity) along the PD axis. $f$ indicates the number of pixels whose intensity is higher than the threshold; $f/fmax$ indicates the ratio between each $f$ and its maximal value along the anatomical axis. **d** Distribution of fluorescence intensity along the PD axis for the indicated conditions. Coloured shading indicates standard deviation, coloured centre indicates mean. Grey shading indicates the standard deviation of the PD position corresponding to elbow (E) or wrist (W). An amputation plane. Source data are provided as a Source Data file.

translocation of cells when overexpressed in the early blastema[9,14]. To test whether this is also true for Tig1, we performed focal electroporations on the distal-most compartment of 4-day blastemas. Concurrent with previous reports[9], transfection of a fluorescent reporter results in the labelling of cells that contribute to distal-most elements, digits and carpals, whilst electroporation of Prod1 leads to the displacement of the labelled cells towards proximal regions, occasionally beyond the amputation plane (Fig. 4a, b). Notably, electroporation of either form of Tig1 results in the proximal displacement of the transfected cells (Fig. 4a, b), with a stronger effect observed for Tig1$^{P155A}$, as more animals exhibited electroporated cells in the proximal segments, including lower and upper arm. These observations show that Tig1 induces proximal displacement of blastema cells that would otherwise contribute to distal domains. In order to track changes in the frequency and distribution of cells along the PD axis, we developed an image analysis algorithm termed Meandros. This

software allows the automated detection and analysis of the distribution of cells within the limb based on their fluorescence (Fig. 4c and Supplementary Fig. 13). Analysis of the scaled limb width indicates that this parameter does not change across different experimental conditions (Supplementary Fig. 14). Quantitative insights into the final location of cells along the PD axis indicate a proximal shift in fluorescence distribution after overexpression of Tig1 and Tig1$^{P155A}$ when compared with RFP (Fig. 4d and Supplementary Fig. 15). While the majority of the cells electroporated with the control plasmid contribute to the distal hand elements, the overexpression of both wild-type and mutant Tig1 leads to a marked extension in the distribution of cells up to the lower (Tig1) and upper (Tig1$^{P155A}$) arm levels, reinforcing the notion that Tig1 overexpression induces proximal displacement.

Meandros analysis highlighted differences in PD distribution between Prod1 and Tig1 overexpressing cells (Supplementary

Figs. 15, 16a). Furthermore, we observed that the combination of Prod1 and Tig1 expression has a differential effect on the extent of proximalisation of either factor alone (Supplementary Figs. 15, 16a). It remains an open question whether Prod1 and Tig1 act independently of each other or they synergise within the same pathway to promote proximal cell translocation. Expression data (see below) suggest that Tig1 acts upstream of Prod1. However, as dose-dependent effects remain possible, this is a notion that merits additional consideration.

While the displacement assay is indicative of the participation of Tig1 in determining the positional outcome of cells, it does not inform on the cellular processes that could mediate this relocation, such as proliferation, apoptosis, migration and/or differences in cell surface tension. To address the contribution of cell proliferation to proximal displacement, we compared the effect of overexpressing Prod1, Tig1 and an anti-proliferative factor with no connection to proximalisation, p53[26]. Whereas all three molecules decrease EdU incorporation upon electroporation in blastema cells (Supplementary Fig. 17a), p53 overexpression does not lead to the proximal translocation of blastema cells (Supplementary Figs. 17b, c, 13, 15), demonstrating that the inhibition of cell proliferation per se is not sufficient to promote proximal displacement, and indicating that additional processes are at play. Given these observations, and considering that we and others[9,14] have not detected any significant changes in apoptosis in the context of proximal displacement, it seems likely that the preferential distribution of cells towards proximal

limb elements is explained by differences in migratory or cell surface tension along the PD axis. Interestingly, Tig1 over-expression negatively impacts the invasive ability of AL1 cells (Supplementary Fig. 18), which could be explained by changes in cell adhesion. Indeed, Tig1 overexpression modulates the expression of adhesion factors (as discussed below) proximally enriched in our analysis (Supplementary Fig. 1c), supporting this hypothesis.

**Tig1 mediates proximo-distal cell surface interactions**. The aforementioned observations led us to functionally test the involvement of Tig1 in mediating proximo-distal differences in cell affinity. To this end, we adapted the blastema engulfment assay[7,11] (Fig. 5a), consisting of juxtaposing blastema mesenchymal tissues derived from similar or different levels along the PD axis and analysing their behaviour ex vivo. Recapitulating the previous observations[7,11], we found that apposing proximal (P) and distal (D) blastemas resulted in engulfment of the distal blastema by its proximal counterpart, while no engulfment took place between blastemas of the same positional value (PP or DD), or between PD blastema pairs treated with anti-Prod1 neutralising antibodies (Fig. 5b, c). To test whether Tig1 mediates such cell affinity differences, we generated custom antibodies against two non-overlapping regions of axolotl TIG1 (Supplementary Figs. 3, 16 and Supplementary Data 5, 6), and addressed the effects of apposing P and D blastemas in their presence.

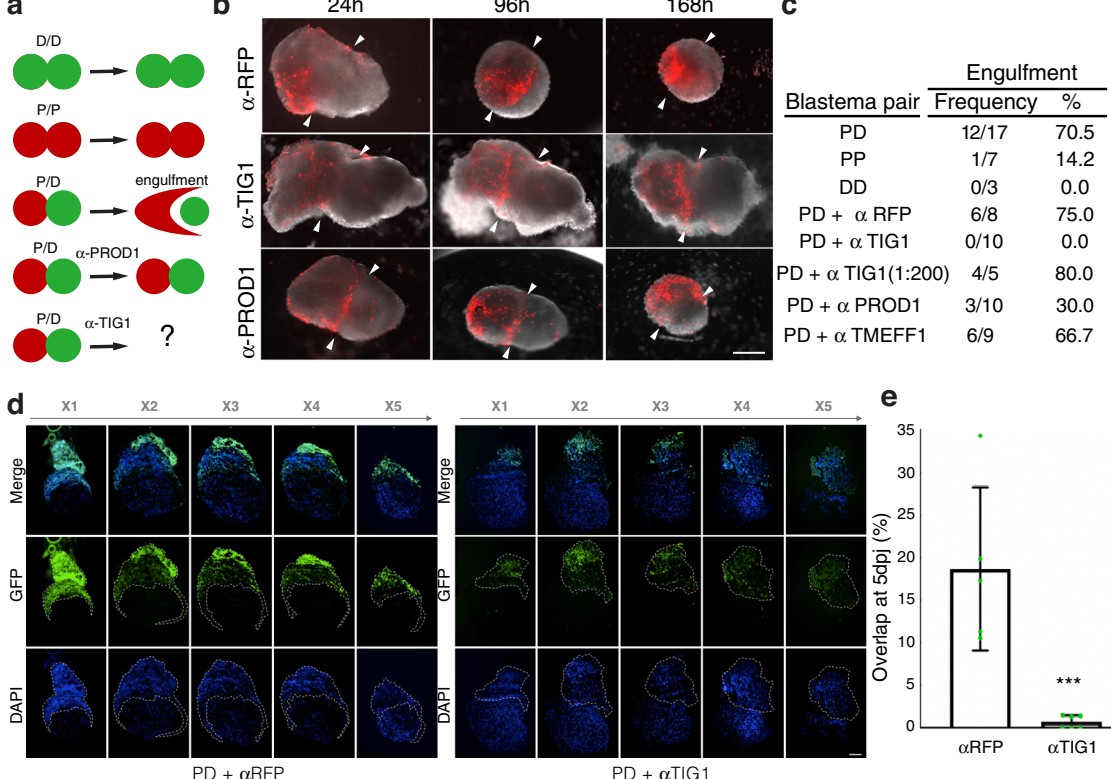

**Fig. 5 Tig1 mediates proximo-distal cell-surface interactions. a** Schematic of the engulfment assay. Juxtaposition of proximal (P) and distal (D) blastemas results in the engulfment of the distal one by its proximal counterpart. **b** Representative images of juxtaposed proximal (red) and distal (white) blastemas at the indicated times post juxtaposition, and after the indicated treatments. Note lack of engulfment following treatment with anti-TIG1 (α-TIG1) antibodies raised against two extracellular epitopes. Scale bar: 500 μm. **c** Quantification of engulfment for the indicated pairs (pooled data from three independent experiments). For α-Prod1, α-Tig1 and α-TMEFF1, [antibody] = 20 μg/ml. **d** Images along an X-stack (x1 to x5) of representative PD blastema pairs at 5 days post juxtaposition (5dpj), following treatment with the indicated antibodies. Proximal blastemas derive from *caggs:eGFP^Etnka* axolotls. Scale bar: 200 μm. **e** Quantification of the overlap between proximal and distal blastemas at 5dpj, based on (**d**) (n = 5 biologically independent samples). ***p = 0.0012 (two-tailed unpaired t-test). Error bars represent SEM. Source data are provided as a Source Data file.

We found that anti-TIG1 neutralising antibodies fully abrogated the engulfment activity, indicating that Tig1 mediates PD differences in cell affinity (Fig. 5b–e). In contrast, treatment with neither a 1:200 dilution of anti-Tig1 antibodies nor with antibodies raised against the extracellular EGF-like domain of TMEFF1 (Tomoregulin)[11], impaired PD engulfment (Fig. 5c). This implies that Tig1 underlies cell surface recognition between proximal and distal cells, providing a mechanistic explanation for its role in proximal displacement.

**Tig1 reprogrammes cells to a proximal identity.** Whether proximal displacement mediated by Tig1 is merely caused by a change in cell affinity or is a consequence of a broader reprogramming of cellular identity towards proximal values is a critical question[9] that remains unanswered. To address this issue, we investigated the impact of overexpressing Tig1 in cells which had already been specified to a distal identity. To exclusively label distal cells, we used a C-Ti$^{t/+}$(Hoxa13:Hoxa13-T2a-mCherry)$^{Etnka}$ reporter line, *Hoxa13:mCherry hereafter*, created by CRISPR/Cas9 technology[27]. In these axolotls, mCherry is fused to the C-terminus of HOXA13, separated by a T2a self-cleaving peptide, and expressed from the endogenous Hoxa13 genomic locus. Validation studies show that mCherry is exclusively detectable in the distal-most part of the limb bud/ blastema during development and regeneration (Supplementary Fig. 19), labelling the field that gives rise to the hand segment in accordance with the model of progressive specification[24,28], thus offering the means to genetically identify blastema cells that have been specified to a distal fate based on their fluorescence. Upon the appearance of the cohort of mCherry$^+$ distal cells in the regenerating limb, we electroporated Tig1 or Tig1$^{P155A}$T2a-GFP constructs, FACS-sorted mCherry$^+$/GFP$^+$ (distal cells overexpressing Tig1) and mCherry$^+$/GFP$^-$ (distal control cells) cells at 3 or 8 days post electroporation, and performed bulk RNA sequencing analysis (Fig. 6a and Supplementary Data 7). PCA analysis indicates clear segregation between control versus Tig1/Tig1$^{P155A}$ overexpressing cells and between timepoints (Fig. 6b). Overexpression of Tig1 versus Tig1$^{P155A}$ has limited impact on clustering, suggestive of phenotypic similarities and consistent with our experimental data. This is also reflected in the differential expression analysis (Fig. 6c). Notably, we found a substantial number of genes (c. 5000) whose expression significantly changed upon Tig1 overexpression ($P < 0.001$), suggesting an extensive transcriptome remodelling (Fig. 6c and Supplementary Fig. 20). Using K-means hierarchical clustering we grouped the top 2000 most variable genes into three main gene clusters (Fig. 6c). Gene Ontology enrichment analysis of the three groups revealed the main GO term categories for each cluster. Cluster A (genes activated by Tig1) is enriched in GO terms related to developmental and tissue morphogenesis processes (Fig. 6d), whereas clusters B (genes activated by Tig1 at all timepoints) and C (genes activated by Tig1 but primarily at 8dpe) are enriched in the immune system and actin cytoskeleton remodelling processes respectively (Supplementary Fig. 21 and Supplementary Data 8).

Critically, in depth analysis of gene expression changes caused by Tig1 overexpression revealed effects in genes related to the establishment of the proximo-distal axis. Tig1 overexpression results in a significant upregulation of the proximalisation factor *Prod1*, a moderate upregulation of the proximal regulator *Meis1*, and the downregulation of *Hoxa* genes, in particular, the distal marker Hoxa13 compared to control cells (Fig. 6e). Consistent with the latter, a network of 11 genes known to be either co-regulated or direct targets of Hoxa13 is also downregulated in response to either of the Tig1 forms (Fig. 6f). Among these are bone morphogenetic protein (BMP) genes *Bmp2* and *Bmp7*,

important for sustaining progressive regeneration and formation of distal elements[28]. In addition, elevated Tig1 levels promote the downregulation of morphogenetic factors associated with blastema growth and patterning, including *Lhx9*, *Lmo1*, *Spry1* and *Lhx2* (Fig. 6g), which are known to be expressed in the distal portion of the developing and regenerating vertebrate limb and are downregulated in the axolotl limb upon proximalising doses of RA[29]. Further, we observed downregulation of *Fgf8* (Fig. 6g), a negative regulator of Meis[30] associated with the specification of the distal limb domain[31]. This inhibitory effect on distal growth provides a molecular explanation for the defects caused by Tig1 overexpression in mid bud blastemas (Fig. 3). Besides its effect on growth factors, Tig1 overexpression also promotes the downregulation of genes encoding factors that promote cell cycle progression, and the upregulation of genes encoding the tumour suppressors *Cdkn1B* and *Retinoblastoma* (Supplementary Fig. 22), consistent with the anti-proliferative effect of Tig1 (Supplementary Fig. 6). Also in agreement with the predicted functions of Tig1, a number of cell adhesion and ECM remodelling factors were found among the differentially expressed genes (Fig. 6h), suggestive of changes in cell affinity and migratory capabilities. These include the downregulation of *Tnc, Fn1, Col3A1, Lsu, Col8A1, Mmp18* and *N-cadherin*, and the upregulation of *Hzal3, Sdc3, Cdh1* and *Mmp13*. Of note, *Mmp13* upregulation has been shown to occur upon proximalising RA doses[29]. Further, *Cdh1* and *Sdc3* were found to be upregulated in proximal connective tissue cells during limb regeneration according to our PCA analysis (Supplementary Fig. 1c). These observations are consistent with the acquisition of proximal ECM and cell affinity expression profiles upon Tig1 overexpression and provide a mechanistic explanation for the effect of Tig1 in the engulfment and displacement assays. Interestingly, whereas our global analysis reveals moderate dynamic changes in gene expression between 3 and 8dpe (Fig. 6b, c), no significant changes are observed for the aforementioned PD-related genes (Fig. 6e–h). This suggests that Tig1 elicits a rapid and stable transcriptome reshaping towards proximal identity in originally distal cells, already manifest at 3 dpe. Altogether, this dataset demonstrates that Tig1 confers blastema cells proximal values and hinders distalisation.

**Tig1 affects the expression of proximo-distal regulators.** In order to validate the RNAseq and obtain further insights into the impact of Tig1 on the expression of PD-related factors, we performed qRT-PCR analysis on *Hoxa13:mCherry* sorted cells after Tig1 electroporation, according to the conditions used for the RNAseq (Fig. 7). We observed that overexpression of either Tig1 or Tig1$^{P155A}$ leads to the upregulation of the proximal factor *Prod1*, as well as the downregulation of *Hoxa11*, *Hoxa13*, the distal growth factor *Lxh2*, and two additional *Hoxa13*-regulated genes, *Enpp2* and *Epha7* (Fig. 7), consistent with the results of our RNAseq analysis. Interestingly, we note that Tig1 overexpression not only leads to an increase in the relative levels of Tig1 mRNA, but also to an increase in Tig1 5'UTR (not present in the pN2-Tig1 vector), indicating that increased Tig1 levels result in the upregulation of *Tig1* transcription, constituting a positive feedback loop (Fig. 7). It is noteworthy that *Meis1* expression shows an upward trend upon Tig1 overexpression even though, due to the high variability between the replicates, the increase does not reach statistical significance. An association between *Meis* and *Tig1* gene expression is also indicated by our PCA analysis of connective tissue cells at 11 dpa (Fig. 1 and Supplementary Fig. 1). Therefore, it is conceivable that *Tig1* activation is achieved through the upregulation of *Meis* factors, given that the Tig1 promoter contains Meis

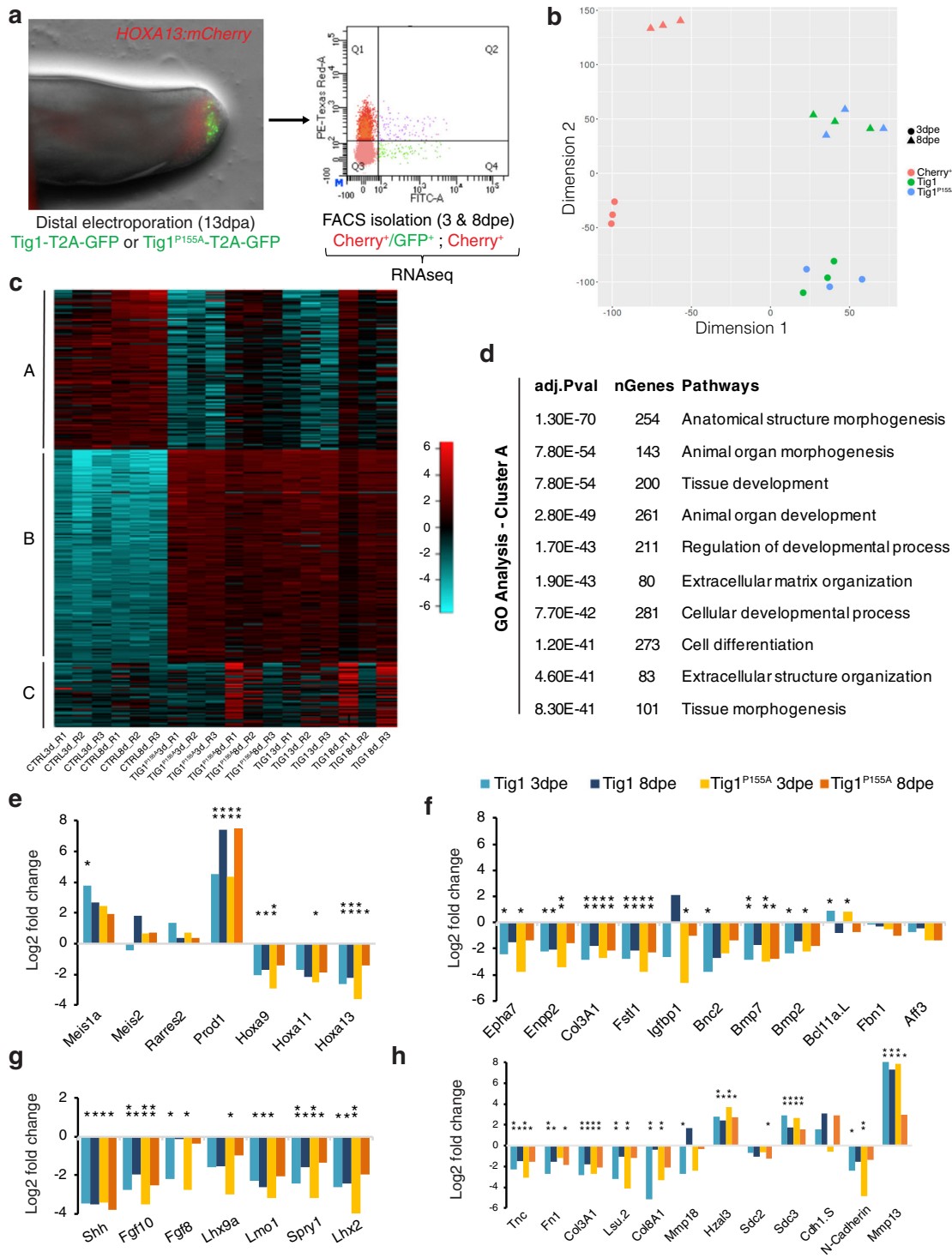

binding sites (Supplementary Data 4). This is a notion that merits further exploration.

Lastly, we asked whether the gene expression signatures associated to Tig1 overexpression match those found in non-perturbed, proximal or distal cells from blastema connective tissue. By evaluating the contribution of Tig1 up or down-regulated genes to the PC1 of early blastema connective tissue cells, we found that most Tig1-upregulated genes contribute to the proximal end of PC1, whereas downregulated genes contribute to the distal compartment (Fig. 8b). Further, we

calculated a score based on Tig1 up or downregulated genes for each cell of the early blastema (Fig. 8a and Supplementary Data 9, 10), which reveals a differential clustering of cells. Cells with a high Tig1-upregulated gene score were found at the proximal side of the UMAP, whereas those with a high Tig1-downregulated gene score were located at the distal-most end. The expression of key PD markers for the top 100 cells of each condition after scoring supports a proximal identity of cells enriched in genes upregulated by Tig1 overexpression, and a distal identity of cells enriched in genes that are downregulated by Tig1 overexpression

**Fig. 6 Tig1 reprogrammes distal cells to proximal identities. a** Experimental design. Tig1-T2a-GFP or Tig1[P155A]-T2a-GFP were electroporated in the distal, mCherry[+] domain of late bud blastemas of *Hoxa13:mCherry*[Etnka] axolotls; FACS isolation followed by bulk RNAseq analysis was carried out for the indicated cells at 3 or 8dpe. **b** tSNE plot using the first and second dimension shows consistent behaviour between replicates and separation between treatments. **c** Heatmap of the 2000 most differentially expressed genes between conditions. **d** GO enrichment analysis of Biological Process for three gene groups determined by K-means hierarchical clustering, displaying the top ten enriched terms for cluster A. The enriched pathways are listed with their corresponding adjusted *p* values and the number of genes. The *p* values are corrected for multiple testing using a false discovery rate (FDR). **e–h** Tig1 overexpression in distal cells promotes a transcriptome shift toward proximal identity. Bars represent ratios of gene expression in Tig1 overexpressing cells versus control cells. **e** Relative gene expression changes in PD markers. Note that Tig1 overexpression leads to the upregulation of proximal factors Meis1a and Prod1 and the downregulation of distal genes including the master regulator Hoxa13 (**e**) and its downstream network (**f**). mCherry protein persists in distal cells even if Hoxa13 is downregulated. **g** Relative gene expression changes in genes associated with blastema growth. **h** Relative gene expression changes in cell adhesion/matrix remodelling factors. Adj. *p* values: *$*p < 0.05$; **$**p < 0.001$. Source data (including exact Adj. *p* values for **e–h**) are provided as a Source Data file.

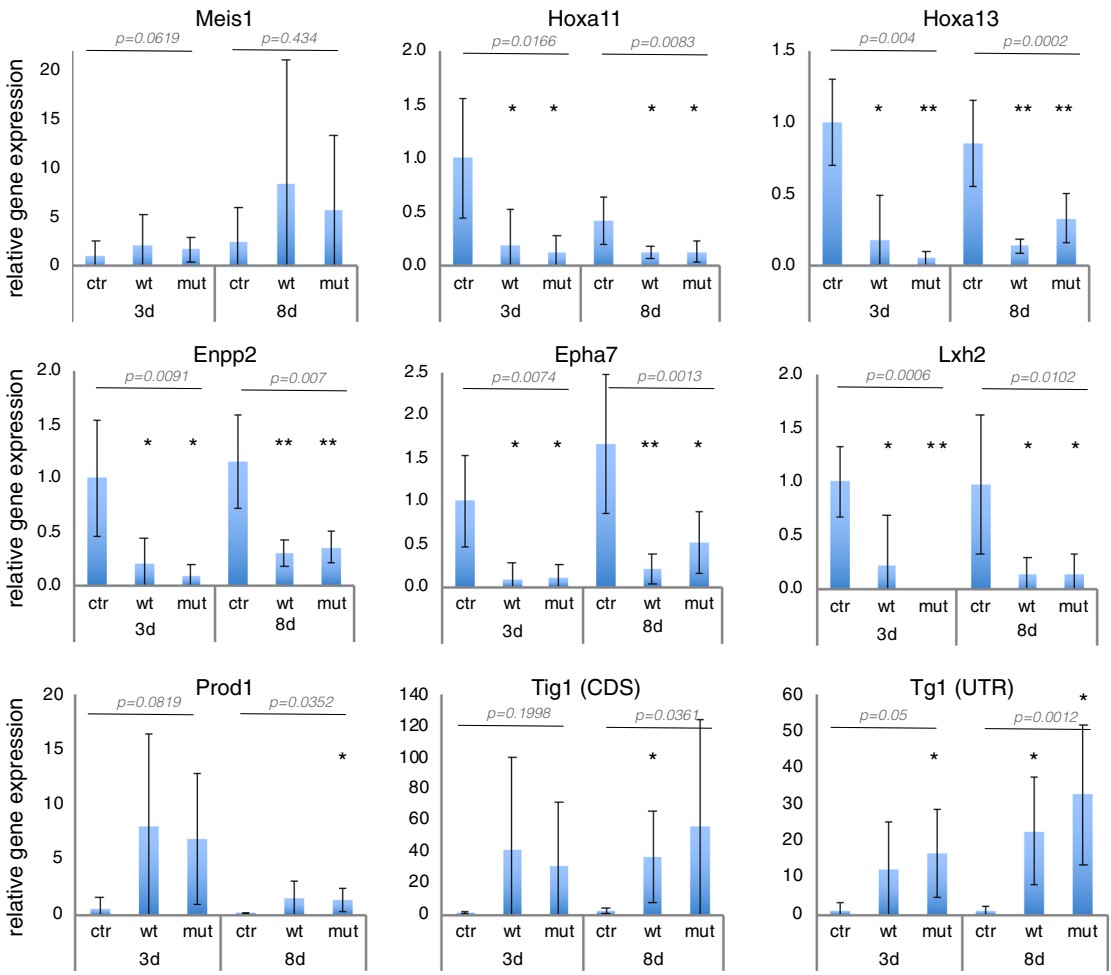

**Fig. 7 Tig1 affects the expression of proximo-distal factors Meis1, Prod1 and Hoxa13.** qRT-PCR quantification of the expression of the indicated factors in FACS-sorted blastema cells from *Hoxa13:mCherry* axolotls at 3 or 8 days post electroporation with the indicated constructs, relative to the control sample at 3 dpe. Rpl4 was used as a normaliser. *p* values are indicated in grey (Welch ANOVA test, group analysis); *$*p < 0.05$; **$<0.001$ (Welch *t*-test for individual analysis, *p* values indicated in the figure). Error bars represent SD. Source data are provided as a Source Data file.

(Fig. 8c). Together, these results suggest that Tig1 is associated with and actively regulates the network of factors that modulate proximo-distal identity in the salamander limb.

## Discussion

The molecular and cellular basis of positional identity remains poorly understood[1,32]. Mechanistic knowledge is currently lacking, from the molecular factors that determine the PD nature of a cell, to their coordination and their re-specification in a regenerative context. Our study offers significant insights into this problem by identifying Tig1 as a determinant of PD identity.

Whilst a system for the specification of PD identity has been proposed to operate at the cell surface level, Prod1 remained its only presumed component. Like Prod1[11], Tig1 is an RA-responsive factor localised to the cell surface and exhibiting graded PD expression in the mature limb. Further, Prod1 expression is regulated by Meis[12] and, as our Tig1 promoter analysis suggests, *Tig1* is also a putative Meis target. Yet unlike Prod1, which is salamander-specific[33] and lacks a membrane anchor in the axolotl[16], Tig1 is highly conserved through

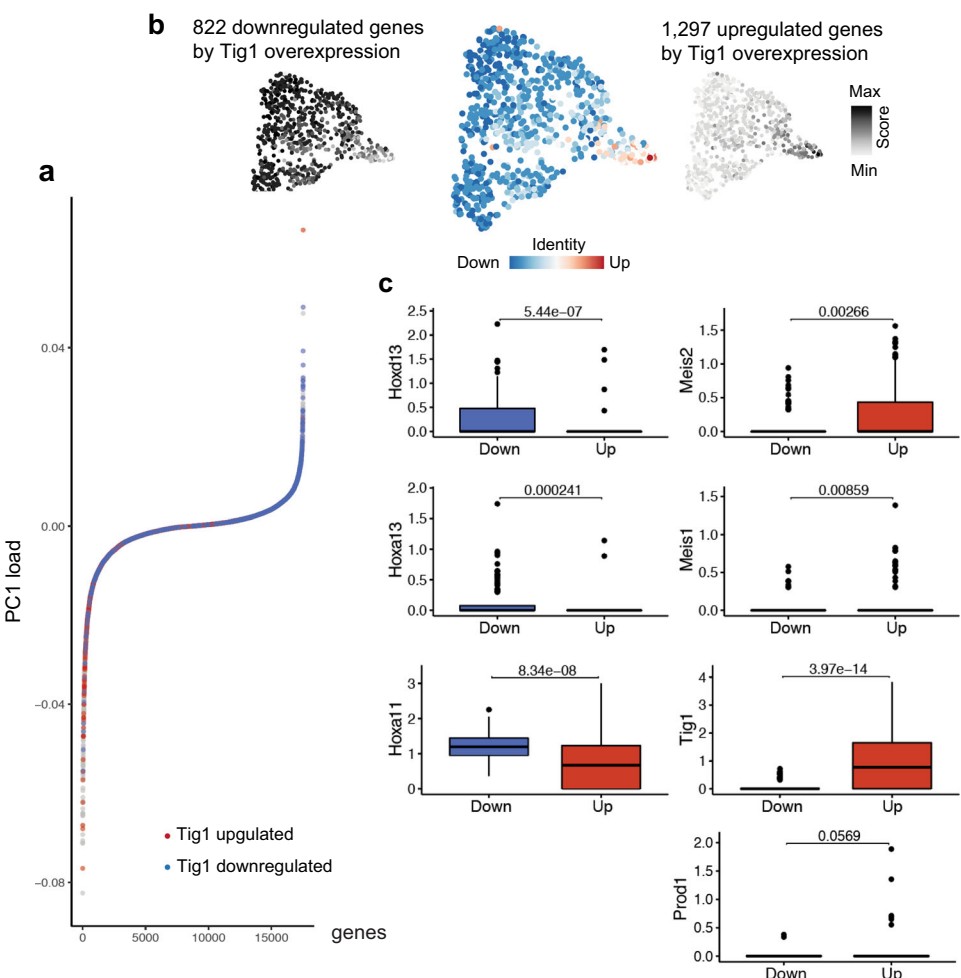

**Fig. 8 Genes up and downregulated by Tig1 map on opposite ends of the proximo-distal axis of the regenerating limb. a** Contribution of up- and downregulated genes by Tig1 overexpression to PC1 (see Fig. 1b). Note that most genes upregulated by Tig1 overexpression map to the proximal portion of PC1, whereas genes downregulated by Tig1 overexpression contribute to the distal portion. **b** Up- and downregulated genes by Tig1 overexpression were used to calculate either an upregulation (right) or a downregulation (left) score for each cell as well. **c** Boxplots represent the gene expression levels for the given genes among the top 100 cells expressing Tig1-upregulated genes (red) and top 100 expressing Tig1-downregulated genes (blue) cells based on the scoring in (**b**). The box horizontal boundaries define the interquartile range (upper boundary: 75th percentile and bottom boundary: 25th percentile of dataset values), the internal line indicates Median, whiskers indicate highest and lowest value for each condition excluding outliers. Results of a *t*-test comparing the gene expression distributions for the two groups are shown above each boxplot panel.

evolution and membrane-bound, raising the possibility that it forms part of a conserved PD determination mechanism. Tig1 resembles Prod1 in that it mediates cell behaviours associated with proximal identity, including the engulfment activity of apposed proximal and distal blastemas, and the displacement of blastema cells to proximal locations. Further, like Prod1[9], Tig1 overexpression leads to patterning defects during limb regeneration, primarily affecting distal elements. Altogether, this strongly suggests that Tig1 is involved in the determination of proximal identity. Yet, our study provides additional evidence for this notion, namely the demonstration that Tig1 is able to elicit the reprogramming of distal cells to a proximal identity.

To our knowledge, this is the first example of a cell surface molecule able to induce a global reprogramming of proximo-distal identity. Based on our observations we hypothesise that the amount of Tig1 molecules at the cell surface determines the expression levels of proximal factors such as Meis1/2 and Prod1 to confer proximal identity. This, together with its graded expression along the PD axis of the mature limb, raises the possibility that Tig1 could form part of a positional identity system encoded at the cell surface level. In this model, Tig1 levels

would confer spatial coordinates to adult tissues and contribute to the sensing of PD positional disparities upon amputation, an essential component of the regenerative response[2,34], as previously proposed for Prod1[11]. It would be of interest to address whether Prod1 and Tig1 act together in order to mediate PD disparity recognition. The existence of cell-autonomously encoded spatial information in adult cells has been well-documented in human fibroblasts, where it is epigenetically maintained[35,36]. It would thus be informative to address if Tig1 is also associated with positional information in mammals, and whether epigenetic mechanisms are responsible for the regulation of its expression as well as that of other PD factors.

Besides the regulation of proximal factors, Tig1-induced reprogramming involves a number of additional, noteworthy processes. First, it entails a marked downregulation of distal factors. Whether the prevention of distalisation is sufficient to proximalise cell identity is an important outstanding question. Second, it is associated with a marked anti-proliferative molecular signature, consistent with the fact that Tig1 counteracts proliferation. It is thus possible that a stable proximal identity is associated with a lack of cell proliferation. This is in agreement

with the decrease in proliferation observed in the proximal-most regions of the late blastema[37]. Importantly, however, suppressing cell cycle progression is not sufficient to promote proximal translocation. Lastly, the mechanisms by which the levels of this cell surface factor determine broad transcriptional changes are as yet unknown. It is conceivable that cytoskeletal arrangements are involved, as Tig1 has been proposed as a modulator of microtubule dynamics[38–40], a notion that should be addressed by further research.

Given the importance of Tig1 for the specification of proximal identity, and taking its plastic responsiveness to RA treatment in both blastema and mature limb tissues into account, we propose that Tig1 is a likely mediator of the RA-dependent proximalisation effect[10,41]. Meis1 has been proven necessary for the proximalising activity of RA in this context[14]. In light of our observations, in particular the upregulation of Tig1 upon RA treatment and the upregulation of Meis1 following an increase in Tig1 levels, forthcoming efforts should identify whether Tig1 is part of the RA-responsive molecular network which regulates proximalisation of regenerating structures.

Together, the body of evidence hereby presented indicates that Tig1 is a central regulator of proximo-distal identity. While this clearly constitutes an important step towards understanding the molecular network underlying PD determination, critical questions remain. These include the nature of the connection between Tig1 and other key factors, such as Prod1 and Meis, in the determination of PD identity, the contribution of Tig1 to a putative system of positional memory, and its mechanism of action at the cellular and molecular levels, particularly whether and how it may act as a sensor of positional disparity between cells. Future studies will provide answers to these questions, delivering fundamental insights into the mechanism underlying positional identity in a vertebrate context.

## Methods

**Animal husbandry**. Procedures for care and manipulation of all animals used in this study were performed in compliance with the Animals -Scientific Procedures- Act 1986 (United Kingdom Home Office), and the laws and regulations of the State of Saxony, Germany. Axolotls (A. mexicanum) were obtained from Neil Hardy Aquatica (Croydon, UK) and from the axolotl Facility at TUD-CRTD Center for Regenerative Therapies Dresden (Germany). Animals were maintained in individual aquaria at ~18–20 °C, as previously described[26]. Axolotls of the leucistic (d/d) strain were used in all experiments, except for the engulfment assay, where Caggs:EGFP[42] transgenics were also used.

**Animal procedures and tissue collection**. Animals of length between 2.5 and 4 cm from snout to the cloaca (that is, between 4 and 8 cm from snout to tail tip) were used in all experiments, except for the engulfment assay, for which animals 7–10 cm from snout to cloaca were utilised.

Axolotls were anesthetised in 0.03% benzocaine (Sigma) prior to limb amputation at the mid-humerus or mid-radius/ulna level as indicated. Animals were allowed to regenerate at 20 °C and limbs or blastemas collected and processed for various assays as specified. Axolotl regeneration stages were defined as previously described[43]. For RA treatments, anaesthetised axolotls were injected i.p. with freshly made stocks of 30 mg/ml all-trans retinoic acid in DMSO, at 100 µg/g body weight. All procedures were performed using Olympus SZX10 microscopes. Injections were carried out with a Narishige micromanipulator connected to a PV820 pressure injector (WPI).

**Tig1 cloning**. For cell localisation studies, A. mexicanum (Am) Tig1 and H. sapiens Tig1 were amplified and cloned into a pEGFP-N2 vector (Clontech) in which the eGFP sequence was removed and a myc tag added in either C- or N- terminal positions, using standard cloning techniques. For expression studies, Am TIG1 or Prod1 were cloned into a pEGFP-N2 vector (Clontech), replacing the EGFP sequence. The p53-N2 vector was previously described (Yun et al. 2013). The Am TIG1[P155A] mutant version was generated using a site-directed mutagenesis kit (New England Biolabs). For baculovirus production, Am Tig1[wt] and Am Tig1[P155A] cDNA sequences were coupled with a T2a element followed by a reporter gene, EGFP or mCherry, and inserted into a pOEMGED baculovirus-rescuing vector (Oliveira et al. 2018). Two fragments corresponding to Tig1 gene and T2a-mCherry sequences were amplified from pN2-TIG1-myc vectors using standard PCR techniques, and integrated into a BmtI and EcoRI-HF-digested

pOEMGED plasmid via Gibson assembly method (NEB), with 1:3:3 plasmid to amplicon ratio. The Tig1-T2a-EGFP cassette was then transferred to a small transfection vector derived from a SacI and SphI-HF – digested pGEM-T plasmid and ligated using classic cloning techniques. The same strategy was used to generate pOEMGED and pGMT-derived vectors containing the Tig1[P155A] sequence.

**Displacement assay**. Forelimbs of 4 cm (snout-tail length) axolotls were amputated at the distal end of the humerus. At 4 days postamputation, the distal region of blastemas was co-injected and electroporated with a reporter plasmid (pEGFP-N2 or pRFP-N2; Clontech) together with the Tig1, Tig1[P155A], Prod1 or a p53 gene plasmids alone or in combinations as indicated (Fig. 3 and Supplementary Figs. 6, 8, 9), using a 1:3 reporter plasmid to plasmid or mix ratio, with a final DNA concentration of 2 µg/µl. Animals were carefully placed in a tightly fit 2% agarose mould in a petri dish with 0.8 PBS, and electroporated using the NepaGene Nepa II machine coupled with CUY611P3-1 electrodes (Xceltis). A bipolar poring pulse of 70 v for 5 ms and subsequent 5 bipolar transfer pulses of 50 v for 50 ms, with an interval of 999 ms and 5% decay were applied. Gene expression was accessed at 1, 7, 12, 28 dpe using light microscopy. Scoring for the localisation of cells according to the upper arm, lower arm and hand levels used to calculate the frequency of displacement phenotypes was done manually. All samples were analysed at the endpoint (28 dpe). This analysis took into consideration any identifiable cell observed at a given PD level.

**Image analysis of cell distribution along the PD axis**. The spatial distribution of electroporated cells and their clones along the proximal-distal (PD) axis of the regenerating axolotl limb was quantified by using Meandros, a Python 3-based image analysis tool that we designed to extract intensity profiles from fluorescent signals spread over non-linear tissue shapes (full methods will be described elsewhere). Through a user-friendly graphical user interface (GUI), Meandros allows simultaneous handling of multiple images, analyse them and run statistical tests on the flight to finally export the analysed data and plots. Briefly, we used the following modules from Meandros: (I) ROI detection: Meandros automatically detected the initial ROI contours delimiting the borders of the regenerating limb from the brightfield of the microscopy images by using a deep neural network approach[44], using the Mask R-CNN solution[45] as architecture and utilising ResNet101[46] as a backbone. When necessary, we fine-tuned this initial automatic guess by using the Ramer–Douglas–Peucker (RDP) algorithm to manually correct the ROI contours and export the corrected limits with Bézier curves. (II) PD axis determination: By using the image brightfield, we identified the PD axis by drawing a Bézier curve crossing the amputation plane, passing by the elbow and the wrist joints together with the digits tip. We arbitrarily settled the amputation plane and the distal extreme of the longest digit tip as 0 and 100% of the PD axis, respectively, and we annotated the elbow and wrist joints as anatomical landmarks for the ulterior analysis. (III) Thresholds and exclusion areas: By using the image channels depicting the fluorescent signal of interest, we selected the image-dependent intensity threshold discriminating from the background the pixels that later were quantified. When necessary, we excluded from the analysis image areas in which unspecific signal was present, such as bone autofluorescence or tissue heterogeneity-derived light diffraction artefacts, by enclosing them within a polygon. (IV) Collapsing 2D images into a 1D axis: We collapsed the 2D image information into the 1D PD axis obtained for each image. To achieve this, Meandros determined the orthogonal direction to the tangent line passing by each pixel of the PD axis. The tangent line in the pixel $i$ from the PD axis was calculated from the least square line fitted to the $i − j$, $i − j − 1$, ..., $i$, ..., $i + j − 1$, $i + j$ pixels, where $j$ was set in its default value of 20. Thus, for each pixel $i$ of the PD axis, Meandros determined the number of pixels belonging to the orthogonal line intersecting the domain within the ROI ($w_i$), and the frequency of them with intensity higher than the previously settled threshold ($f_i$). Observables: We quantified each PD profile as the spatial distribution along the PD axis coordinate $i$ of the ratio between each $f_i$ and its maximal value along the PD axis $f_{max}$ ($f_i/f_{max}$). We also quantified the local limb width profile as each $w_i$ along the coordinate $i$ of the PD axis.

**Engulfment assay**. To generate blastemas of distinct proximal-distal values, forelimbs of 8.5 cm (snout-vent length) animals were amputated either at the stylopod at the distal end of humerus, or at the distal end of radius/ulna level. At 12 days postamputation, blastemas along with 2 mm of the adjacent stump were collected and washed twice in A-PBS (80% 1X PBS). The epidermis was carefully removed with the help of fine forceps.

When using exclusively DD white axolotls, one blastema of each pair was labelled with Fast-DiO or CM-DiI at a concentration of 0.15 mg/mL and 0.25 mg/ml respectively, diluted in serum-free MEM and incubated for 30 min at 25 °C, and finally washed twice with A-PBS. In the other cases, the pairs consisted in one white and one GFP (Caggs:EGFP[42]) -derived blastema.

The stump segment was removed, thereby creating a basal surface corresponding to the proximal-most area of the blastemas. Blastema pairs were then transferred to wells of 24 or 96 multi-well plates and juxtaposed at their basal surface. Space in between wells was filled with water to maintain a moist environment, and the plate was incubated at 25 °C for 20 min. Blastemas retain a

thin PBS layer on its surface that prevents them from drying for such a short period of time. MEM supplemented with 5% FBS was carefully added to the wells to cover the blastema pair, and incubation proceeded for 7 days at 25 °C, 5% $CO_2$. Images were acquired in Zeiss Axio Zoom V16 microscope (Zeiss) and processes using Zen 2.3 (blue edition) software.

Treatment with antibodies against *Am* TIG1, *Am* PROD1, *Am* TMEFF1 or control IgG rabbit antibody (antibody section) was performed at a final concentration of 20 µg/ml in MEM supplemented with 5% FBS. The solution was added to the juxtaposed blastema pairs after a 20 min incubation period and maintained until the end of the experiment.

**Baculovirus-mediated transduction.** Baculovirus-mediated transduction of axolotl tissues was performed as described elsewhere[25]. Briefly, Baculovirus was pseudotyped with *vsv-ged* gene, inserted into the rescue vector under the baculovirus polyhedrin promoter. Genes of interest (Tig1wt-T2a-EGFP and Tig1P155A-T2a-EGFP) were cloned into the baculovirus rescue vector under the *CMV* promoter using standard restriction enzyme methods. Pseudotyped baculoviruses carrying the gene of interest were generated by co-transfection of the rescue vector together with replication-incompetent baculovirus DNA into a modified *Spodoptera frugiperda* cell line (expresSF+, ProteinSciences Corporation). Upon culture expansion, recombinant baculoviruses were collected, concentrated and purified. Baculovirus titre was assessed by end-point dilution assay in SF-9 Easy Titer cells, a modified *Spodoptera frugiperda* cell line expressing a reporter *egfp* gene under the control of the baculovirus polyhedrin promoter.

Expression of the genes of interest in axolotl blastemas was achieved by injecting the corresponding baculovirus (2 µl each; Bv titre: c. $1 \times 10^{12}$) into the upper arm blastemas as indicated. Injections were performed using a glass capillary needle pulled on a micropipette puller (Sutter Instrument Co. model P-97). Glass capillaries (1.2 mm OD, 0.9 mm ID borosilicate glass) were acquired from Harvard Apparatus.

**Bone and cartilage staining and segment measurements.** Regenerates were fixed with 1% MEMFA, washed with PBS-Tween (0.2%), equilibrated for 10 min in 25 and 50% EtOH, and then incubated in alcian blue-60% EtOH solution for 1 h at 37 °C. Samples were washed for 1 h in 60% EtOH–acetic acid solution, placed in 95% EtOH for 15 min and then in 1% KOH solution for 10 min, followed by a 35 min incubation with alizarin red-1% KOH solution at room temperature. Samples were washed in 1% KOH–85% glycerol for 10 min and then overnight, and stored in 20% glycerol–PBS. Images were acquired and the length of the upper arm, lower arm and hand segments was estimated by measuring the length of the humerus, radius and ulna, and from the wrist until the tip of the second finger, respectively.

**Cell culture.** Axolotl AL1 cells (initially derived by S. Roy, University of Montreal, Montreal, Canada) were grown on 0.75% gelatin-coated plastic dishes in MEM (Gibco, UK) supplemented with 10% heat-inactivated foetal calf serum (FCS, Gibco), 25% $H_2O$, 2 nM L-Glutamine (Gibco), 10 µg/ml insulin (Sigma, St Louis, MO) and 100 U/ml penicillin/streptomycin (Gibco) in a humidified atmosphere of 2.5% $CO_2$ at 25 °C. Cell subculture was performed as previously described[47]. 293 T cells were grown on plastic dishes in DMEM (Gibco) supplemented with 10% heat-inactivated foetal calf serum (FCS, Gibco), 2 nM L-Glutamine (Gibco) and 100 U/ml penicillin/streptomycin (Gibco) in a humidified atmosphere of 2.5% $CO_2$ at 37 °C. Retinoic acid (RA; all-trans retinoic acid, 50 mg/ml stock solution, Sigma) treatment was performed as described[48].

**AL1 cell electroporation.** AL1 cell suspensions ($10^5$ cells in 0.1 ml) were mixed with 2 µg of the indicated constructs (pN2-amphiregulin-myc, pN2-myc-amphiregulin, pN2-Tig1-myc and pN2-myc-Tig1) and nucleofected using a Nucleofector II (Lonza) as per manufacturer's instructions. The nucleofected cells were added to 1.5 ml supplemented minimal medium (MEM, Gibco) and incubated at 25 °C for 20 min prior to plating into gelatin-coated dishes. Expression of myc-tagged Tig1 was confirmed through Western blot analysis, as previously described[26]. Live staining was carried out at 48 h post nucleofection.

**293T and AL1 lipofection.** Cells were grown to 70% confluency and lipofected using lipofectamine 2000 (Thermo Fisher). Briefly, 8 µl Lipofectamine solution were mixed with 200 µl OPTI-MEM (Gibco), incubated for 5 min at room temperature and added to a pre-mix of 3 µg of the indicated constructs plus 200 µl Opti-MEM. After a 20 min incubation, the 400 µl solution was added to the cells, in their own growth media (1.5 ml), drop-by-drop. Cells were incubated for 24 h prior to live staining. Volumes of lipofectamine, OPTI-MEM, penicillin-streptomycin-free media and the amount of DNA constructs lipofected were scaled according to the culture format.

**Scratch assay.** AL1 cells were seeded at 80% confluence into Ibidi culture inserts and lipofected overnight. Subsequently, the culture inserts were removed (leaving a cell-free gap 500 µm in width), and cells were cultured in control or antibody-supplemented complete media. Cells were imaged every 24 h and gap closure was

quantified as the % cell-free area remaining relative to the 0 h timepoint. The location of the culture insert was marked prior to removal to ensure the same field of view was imaged at each timepoint. For antibody treatments, AL1 cells were treated with a combined concentration of 2 µg/ml of equal parts of α-TIG1 antibodies, or α-RFP IgG rabbit antibody, in complete media.

**qRT-PCR.** RNA was isolated from axolotl AL1 cells or indicated tissues using Tri Reagent (Sigma) and randomly primed cDNA synthesised using Superscript II (Invitrogen). Gene expression was determined by quantitative real-time PCR using the indicated primers (Supplementary Table 1). RT-PCR was carried out using iQ SYBR Green supermix (Bio-rad, Hercules, CA) on a Chromo 4 instrument running Opticon 3 software (Bio-rad). All reactions were run in triplicate and at least three independent RNA preparations were analysed for each sample.

Axolotl upper arm blastema including 500 µm of tissue proximal to the amputation plane from 4–4.5 cm nose-to-vent long animals were collected in three or four biological replicates. RNA was purified by Qiagen RNeasy mini kit and poly-dT primed cDNA was synthesised from 100 ng total RNA using Superscript III (Invitrogen). For Tig1 expression, qPCR was measured twice using two different primer pairs; one within the CDS and one within 5′UTR. Tig1 expression was normalised to the expression of the ribosomal protein Rpl4. All conditions were normalised to the mean of the intact limb levels. Statistical significance between timepoints was calculated by two-way ANOVA test using timepoint and probe as factors, assuming variances were not equal. Adjusted *p* values were calculated using Dunnett's multiple comparison test.

For newt tissues, samples were collected from 3 to 4 years old *Notophthalmus viridescens* newts. Norgen RNA extraction kit was used for RNA isolation and cDNA was prepared using Superscript-IV kit. Expression levels relative to ef1α (Supplementary Table 1) were calculated using the Livak $2^{-\Delta\Delta Cq}$ method.

For FACS-sorted blastema cells: three biological replicates per condition of the libraries used for the RNA-seq experiment plus three additional replicates, prepared in the same way (see below), served as basis for the qPCR to confirm gene expression changes caused by Tig1 overexpression. The minimum number of biological replicates for each condition was 5. qPCR was carried out using iQ SYBR Green supermix (Bio-rad, Hercules, CA) on a Chromo 4 instrument running Opticon 3 software (Bio-rad). Gene expression was normalised to the ribosomal protein Rpl4. For visualising individual genes, all conditions were normalised to the mean value of the control condition at 3 dpe.

**EdU incorporation assays.** To determine the percentage of cells in the S-phase, AL1 cells were sub-cultured and incubated in a growth medium supplemented with 5 µM 5-ethynyl-2′-deoxyuridine (EdU) for 24 h. Cells were then fixed in 4% PFA for 10 min and stained using Click-iT Edu Alexa Fluor 594 Imaging kit (Life Technologies) according to the manufacturer's instructions.

To detect EdU incorporation in salamander tissues, 10 mM EdU (20 µl per animal) were administered by i.p. injection at 24 and 48 h after electroporation. Blastemas were collected 72 h post electroporation, fixed O/N in 4% PFA, embedded and cryosectioned. EdU incorporation was determined using Click-iT Edu Alexa Fluor 488 Imaging kit (Life Technologies) on tissue sections. Estimation of EdU incorporation was performed by calculating the number of EdU+ cells over the total number of transduced cells (RFP+).

**Generation of C-Tit/+(Hoxa13:Hoxa13-T2a-mCherry)Etnka transgenic axolotls.** The endogenous HOXA13 protein was tagged at the C-terminus according to the published protocol[27]. Briefly, a portion of the Hoxa13 gene including a part of the single intron and the remaining downstream part of the CDS was amplified from the axolotl genomic DNA and inserted into the vector pGEM-T along with a DNA fragment encoding T2a-mCherry-3xnls using Gibson assembly cloning (BioLabs). The resulting vector was injected into fertilised eggs along with the CAS9 protein (MPI-CBG) and the gRNA targeting the sequence GGGGGCTTTTGCGGGTTTTCC, which is present in the intron sequence within the vector and in the genome. The insertion is mediated by NHEJ and is thus expected to cause a small InDel within the intron, but it will result in a seamless Hoxa13 CDS. T2a-mCherry is followed by the pGEM-T poly(A) signal and the rest of the vector.

Validation: Tit/+(Hoxa13:mCherry) axolotls exhibit expression of mCherry in the distal part of the developing limb bud throughout limb formation (Supplementary Fig. 10). During regeneration, mCherry expression is observed in the distal portion of the blastema from mid bud onwards, coinciding with the appearance of the Hoxa13 domain[24] and remaining until the digit stage. The animals used in the experiment were heterozygous for the T2a-mCherry tag, to overcome any potential loss of function that could be originated from the knock-in.

**Antibodies.** Polyclonal anti-*Am* TIG1 antibodies were manufactured by Eurogentec, by raising rabbit antisera against two non-overlapping peptides corresponding to residues MAQVKSVKQRLRND (154) and GSSGGRSEEGSASF (155) of the full-length axolotl TIG1 protein, using KLH as a carrier. Both reagents were analysed by solid-phase ELISA and affinity purified using their respective immobilised peptides (Supplementary Data 2, 3). IR/Odyssey-based Western blotting was performed as previously described[26]. Note that while these reagents work for Elisa and western blotting, we did not manage to find conditions suitable for

immunofluorescence applications. The two affinity-purified anti-PROD1 anti-bodies were generated as previously described[11]. Anti- axolotl PRRX1[17] (kind gift from Prayag Murawala, final concentration 200 ng/ml), Anti-Myc (mouse mono-clonal 9E10, Sigma-Aldrich, 1:500) and anti-RFP (Rockland, 1:200) were also used in this study.

**Live-cell staining and immunofluorescence**. For live-cell staining, 293 T or AL1 cells were incubated for 3 h with anti-Myc mouse monoclonal antibody (Sigma-Aldrich; 1:500), and secondary staining was performed using anti-mouse AlexaFluor568 antibodies (Invitrogen; 1:1000). Hoechst 33258 (2 μg/ml) was used for nuclei counterstaining. Samples were observed under a Zeiss Axioskop2 microscope and images were acquired with a Hamamatsu Orca camera. Whenever comparative analyses between samples were performed, all images were acquired with identical camera settings and illumination control. Image processing (contrast enhancement) was equally applied to all matched experimental and control samples using Improvision Openlab (PerkinElmer) software.

**Tissue dissociation for RNAseq**. Limbs of Hoxa13:Cherry transgenics (4.5 cm snout-vent length) were amputated at the stylopod level, near the elbow. The resulting blastemas were electroporated with Tig1-T2a-GFP or Tig1[P155A]-T2a-GFP plasmids at 13 dpe, and harvested at 3 or 8 days post electroporation. Twelve blastemas were pooled per condition, washed in 0.8 PBS, and incubated in 500 μl of 1x Liberase[TM] research-grade (Sigma-Aldrich) for 5 min at room temperature. The epidermis was removed and the blastema mesenchyme dissociated with the help of very fine forceps. The incubation proceeded at room temperature for 30 min with a rocking motion at 40 rpm. Cells in solution were gently pipetted up and down, filtered through a 70 μM Filcon (BD™ Medimachine) and collected into a falcon tube. Sytox blue (Thermo Fisher) was added at 1:2000 and the solution was kept in ice until FACS sorting (Supplementary Data 11). After sorting, cells were kept in 1x freshly-prepared lysis buffer containing RNAse inhibitor and kept at −80 °C. cDNA library preparation from 300 cells per sample (three replicates per condi-tion) followed by SMARTseq2 sequencing was performed by CMCB-Deep sequencing facility/Dresden-concept Genome Center.

**Bioinformatic analysis**

*In silico modelling of the proximo-distal axis from scRNAseq data*. Single-cell RNA-seq data from genetically labelled (via LoxP-reporter and Prrx1 limb enhancer driving expression of the Cre recombinase) FACS-sorted connective tissue cells from 3, 5, 8, 11 and 18 dpa limb blastemas[17] were analysed as follows: gene expression (log2(transcripts per million)) was used as input for principal compo-nent analysis. The top twenty loadings of the positive and negative axes of the first nine principal components were inspected to identify principal components that segregated proximo-distal markers on separate ends (Supplementary Data 1). Principal components 3, 5 and 9 in timepoint 11 dpa had the largest distance between Meis2 (proximal) and Hoxa13 (distal), followed by PC6 and PC3 of 18 dpa. We therefore adopted PC3 of 11 dpa as a model for the proximo-distal axis based on single-cell RNAseq data.

Subsequently, a recently published single-cell RNA-seq dataset of axolotl blastema formation (GSE165901)[19] was re-analysed. Only the blastema collected 11 days after amputation yielded enough cells with a descent expression of well-established markers (Meis1, Meis2, Hoxa13 and Hoxd13) for the proximal-distal axis. The Seurat package (v4)[49] was used for almost all analyses and visualisations unless stated differently. Cells with a percentage of mitochondrial genes less than 10%, and over 500 but under 30,000 RNA counts, were selected. A principle component analysis was performed and the first 50 PCs were used as input for Louvain clustering (resolution of 0.8) and dimensionality reduction by a UMAP embedding. Connective tissue clusters were identified by the canonical markers Prrx1 and Col6A1 and extracted ignoring pericytes expressing Myh11 and other CT clusters that are solely characterised by low RNA counts. Obtained cells were re-clustered using the same parameters and remaining doublet clusters with epidermis cells expressing EPCAM or macrophages expressing C1QB were further filtered out. The final dataset yielded 810 pure CT cells. Cell cycle scores were obtained and regressed out together with differences in RNA/Gene counts and the percentage of mitochondrial genes during data scaling with Seurat. After running a new PCA (Fig. 1b) for the final dataset, the first 10 PCs were used as input for Louvain clustering (resolution 0.3) and UMAP embedding (Fig. 1a). Feature plots (Fig. 1c) were generated by calculating feature-specific contrast levels based on 5 and 95% quantiles of non-zero expression. Seurat's AddModuleScore function[49] was used to calculate a distal score based on Hoxa13 and Hoxd13 expression and a proximal score based on the expression of Meis1, Meis2 and Prod1 with five controls per bin, respectively. The scores were united by subtracting the distal score from the proximal score (Fig. 1d). A cutoff of 0.3 and −0.3 for the combined score was applied to assign a proximal or a distal identity to the cells, respectively. Afterwards, the FindMarkers function was used to identify genes that are differentially expressed between the distal and proximal cells based on a Wilcoxon rank-sum test (Supplementary Data 3). The result was visualised as a volcano plot (Fig. 1e) generated with the 'EnhancedVolcano' R package[50]. Genes down- or upregulated by Tig1 overexpression (Supplementary Data 7) were intersected with the genes detected in the scRNAseq dataset which resulted in 822 downregulated

and 1297 upregulated genes that are reliably detected (sum of normalised counts across all cells must be greater than 10) in the scRNAseq dataset. The AddModuleScore function was again used to assign scores based on the gene lists obtained with 100 control genes per bin, respectively. As described above the scores were united (Fig. 8) and the resulting top 100 'proximal' and top 100 'distal' cells were extracted to visualise the gene expression of important proximal-distal marker genes as boxplots by using the R package 'rstatix' and 'ggpubr' (Fig. 8).

*Tig1 promoter analysis*. A putative Tig1 promoter region corresponding to 40Kb upstream the start site (axolotl-omics.org) was analysed using the web interface https://molotool.autosome.ru/.

*RNAseq reference sequence and annotation*. Sequence and gene annotation of the axolotl nuclear genome assembly AmexG_v6.0-DD were downloaded from https://www.axolotl-omics.org. The gene models for the HoxA and HoxD genes were replaced with manually curated gene models (personal communication with Sergej Nowoshilow, axolotl-omics.org). Sequence and gene annotation of the mitochon-drial genome assembly (NCBI GenBank, AY659991.1) and RNA spike-in control sequences (ERCC, Ambion; for quality control) were included as well.

*RNAseq mapping and counting*. RNA-seq reads were mapped against the reference using STAR (v2.7.6a; PMID:27115637) and splice sites information from the gene models. Uniquely mapped reads were converted into counts per gene model and sample using featureCounts (v2.0.1; PMID:30783653).

*DGE analysis*. Differential gene expression analysis was performed using DESeq2 package[51]. The iDEP.93 web application[52] was used for K-means clustering and g:profiler (https://biit.cs.ut.ee/gprofiler/gost) for GO enrichment. From the initial 44224 genes in 18 samples, 19761 genes passed the filter (minimum 0.5 counts per million in at least one sample). K-means clustering across all samples was per-formed using the top 2000 variable genes as ranked by standard deviation. The number of clusters, k = 3 was chosen based on the elbow and tSNE method. Pathway enrichment analysis for each cluster was conducted based on Gene Ontology (GO) Biological Process database[53]. P values are corrected for multiple testing using false discovery rate (FDR).

**Statistical analysis**. Animals in each sample group were randomly selected. Sample group size (*n*) is indicated in each figure legend, while all experiments were carried out in at least three biological replicates. Statistical analyses were performed with Prism 4.0 software. The method used for each analysis is indicated in the corresponding section.

**Reporting Summary**. Further information on research design is available in the Nature Research Reporting Summary linked to this article.

## Data availability

All data generated or analysed during this study are either included in this published article (and its supplementary information files) or publicly available[17,19]. The raw RNAseq data generated within this study have been deposited in the database under the Gene Ontology Omnibus (GEO)/NCBI database under accession code GSE184948. scRNA seq and bulk RNAseq analysis, as well as Tig1 antibody generation and validation information, is presented as Supplementary Material. Source data are provided with this paper.

## Code availability

Access to all custom-generated code pertinent to this manuscript is available upon request.

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

## Acknowledgements

We thank all members of the Yun Lab for critical comments, Jifeng Fei for advice on knock-in transgenesis, Jeremy Brockes for insightful discussions, Gabriel Waksman for institutional support, MPI-CBG Protein Expression Facility for protein/viral expression support, TUD-CMCB Histology, Flow Cytometry and Light Microscopy Facilities for tissue processing and imaging support, DRESDEN-concept Genome Center for RNA sequencing, and Beate Gruhl, Anja Wagner and Dominick Kruger for animal care. A.E. was supported by a NIH Ruth Kirschstein postdoctoral fellowship (F32GM117806), H.E.W. by an Alexander von Humboldt postdoctoral fellowship, H.A. by a CONICET-UADE scholarship, A.P. by a Deutsche Forschungsgemeinschaft grant (INST 269/768-1), O.C. by CONICET (Argentina) and ANPCyT grants (PICT-2014-3469 and PICT-2017-2307), E.M.T. by ERC Advanced grant (294324) and DFG grant (TA274/3-3); A.S. by Cancerfonden, Swedish Research Council, Knut and Alice Wallenberg Foundation, Stiftelsen Olle Engkvist Bryggmästare, and M.H.Y. by Deutsche Forschungsgemeinschaft grants (DFG 22137416 & 450807335) and TUD-CRTD core funds.

## Author contributions

C.R.O., D.K., S.G.G.M., P.B.G., H.E.W. and E.S. performed experiments; C.R.O., S.G.G.M., A.E., D.K., T.G., P.B.G., H.E.W., A.P, H.A., R.C.C., E.S., O.C., A.S. and M.H.Y. analysed and discussed data; C.R.O., D.K., S.G.G.M., P.B.G., H.E.W. and M.H.Y. designed experiments; E.M.T. contributed reagents; M.H.Y. supervised the project and wrote the manuscript with contributions from all authors.

## Funding

## Competing interests

The authors declare no competing interests.
