## [Peer Review File · Nature Communications]

Tig1 regulates proximo-distal identity during salamander limb regenerationReviewers' Comments:

Reviewer #1:

Remarks to the Author:

This is a nicely written paper that describes a new protein candidate for proximal positional identity within the axolotl salamander arm. Previously, Prod1 was identified as a proximal positional identity protein in the newt and axolotl. Data are presented in this paper to argue that Tig1 also satisfies criteria of a proximal positional identity protein. Tig1 is clearly an interesting molecule and appears to affect aspects of cellular identity and limb regeneration given the experiments that were performed, and importantly results from this study further implicate a relationship between cell proliferation and proximalization. However, the connection between Prod1 and Tig1 in determining proximal identity remains unknown and some issues need to be clarified/addressed in the manuscript to better understand the significance of the work and whether or not it will move thinking beyond what has already been established from studies of Prod1.

Comments/Concerns

P2, Line 34. Does mature limb mean intact limb? This is a bit confusing to me and perhaps others that are not in the limb field. It should probably be noted here that previous studies have not found a graded expression pattern of Meis expression in the mature/intact limb. Doesn't this contradict the principles that are proposed for a PD molecule and coordinate system?

P2, Line 35. Need to make it clear here that Prod1 with a GPI anchor was originally identified in the newt. Without this qualification it is confusing when the axolotl GPI-anchorless version is introduced.

P3, Line 22. This wording seems a bit strong... "that accurately modelled changes". A static principle component/variable calculated for a single sample/time point cannot model changes, whether they be accurate or inaccurate.

P3, Line 24. Please state the total number of cells that were positive (relative to a threshold) for Hoxa13, Meis1, and Tig1. It seems that the resolution PC3 provides is minor (how much variance explained) and mostly associated with low Hoxa13 expression.

P3, Line 24. The Figure 1 legend says that Prod1 expression is not available in the single cell dataset. Might Prod1 align to "AC_02200036472.1 in the transcriptome that was used to annotate the single cell data and if so are there expression data for this transcript model? If not, and Prod1 is expressed by blastema/connective tissue cells, how was it missed in the single cell study that enriched for connective tissue cells? It seems from other single cell data that Prod1 might be expressed highly in immune cells and less so in connective tissue cells. Might Prod1 expression associate with immune cells and not connective tissue cells or perhaps an immunological cell phenotype? It would seem important to show in situ data for Prod1 and Tig1 in the manuscript as the proposed PD coordinate system assumes both molecules contribute in some unresolved way.

P3, Line 46. It is known that aneuploid AmAL1 cells express Prod1 and Meis1/2. Is this true for NvA1 cells?

P4, Line 24. Not clear the data support the idea of strong Tig1 downregulation when there is unequal sampling of early vs late timepoints. With more late samples one might conclude there is a gradual decrease in transcripts.

P4, Line 37. It's not clear that the following conclusions - Tig1 establishes PD patterning and Tig1 is strongly implicated in determination of proximal identity - follow from the misexpression results. The data in this paragraph show a disruption of limb regeneration, which could trace to regulation of cell proliferation, which was shown in the preceding paragraph, or disruption of some other biological

process.

P5, Line 19. Why is there such a big difference in how Prod1 vs Tig1 expression displaces cells? Could Prod1+ cells include non-blastema migratory cells? I don't really see an additive effect of Prod1 and Tig1, if anything, the combination of the two seems to reduce Prod1's overall effect. Perhaps this should be quantified to strengthen the argument?

P5, Line 25. These cell proliferation data do not provide much further resolution. Other important processes – migration and cell adhesion – are mentioned to be important but no data are provided.

P6&7. The gene expression data should be made available as a supplementary file showing p values and fold change values. What criteria specifically were used to identify the top 2000 genes? The caption and Figure 6 legend should probably indicate that Tg1 overexpression affects the expression of the indicated genes, not controls their expression.

P8, Line 42. The proposition that Tig1 is a mediator of RA-dependent proximalization could be investigated further by comparing transcriptional changes mediated by Tig1 over-expression to those mediated by RA administration.

Reviewer #2:

Remarks to the Author:

This is an exciting manuscript by Oliveira et al presenting evidence that the molecule Tig1 is important in determining the positional information of fibroblasts during salamander limb regeneration. How positional information is stored and utilized in salamander limbs in order to regenerate the appropriate proximodistal structures is a longstanding and important question in the field.

Noteworthy results of the manuscript include the identification of a new molecule that can change the positional information of limb cells in a regenerating salamander limb. Few other molecules (prod1 and meis1 and 2) have been shown to have similar properties so the identification of a new molecule is noteworthy. As a side note, it is unclear how many genes may generate similar phenotypes. The work will have considerable impact in the field of limb regeneration and I assume that many others studying limb development will also take notice. Discovery of the previous molecule, prod1 (or C59), has had a lasting impact on the field of regeneration biology, where people have studied its role in a number of species including lizards and zebrafish appendage regeneration. How positional information is maintained and established during limb regeneration is still a very open question and is of critical importance for our understanding of regeneration. Although the experiments presented here are classic in the field, the novelty is in the molecule identified.

The experiments do support the conclusions although the experiments are slightly limited, likely due to limitations with the current methods used in the animal model and the significant amount of work it takes to generate genetically modified axolotls. The study meets the standard in the field (axolotl), although the phenotypes presented are less severe than expected, which may be due to the limitations of electroporation and the new techniques of baculovirus rather than the function of the gene of interest. The engulfment assay phenotypes are also somewhat limited compared to sorting assays that are performed in chicks and mice limb bud experiments. These are more limitations of the current methodologies used in axolotls rather than the approaches used in the study. Overall, I applaud the use of in vivo, ex vivo, and in vitro functional approaches to address the problem. The authors used the appropriate techniques available in the animal model to address the question.

Although the gain of function experiments are overall convincing, I'm missing the spatial expression profiles of prod1 and tig1 in the limb. Is Tig1 expressed in all connective tissue fibroblasts? RNAseq isn't enough for me considering only prrx+ cells were studied. Also, the expression pattern of prod1

hasn't really been shown with modern FISH techniques. I think this is a general weakness of the manuscript.

In relation to the above statement, I suggest modification in the writing, especially in the introduction, to discriminate between experiments performed in axolotls versus newts. The authors build the story as if it is all cohesive between species. Much of the early work on prod1 was done on newts, and much of the grafting work on axolotl. Considering the differences in prod1 localization (cell surface versus secreted), differences on muscle regeneration (dedifferentiation versus stem cell based), and differences in the impact of denervation (permanent inhibition versus temporary), it is becoming less convincing to extrapolate results in one species to the other.

This point is especially true in the following statement at the end of the introduction: "This raises the possibility that, in order to achieve a graded distribution along the PD axis and execute its functions, Prod1 interacts with additional, as yet unidentified components of the proximo-distal memory system."

Several studies contradict this statement in axolotls. Prod1 was higher in distal blastemas compared to proximal blastemas according to McCusker et al., 2015, prod1 did not respond to RA treatment in Nguyen et al., 2020 (I just analyzed the data on NCBI GEO database to doublecheck), and did not show up as differentially expressed between proximal and distal samples in Bryant et al 2017 or Voss et al 2018. Although each of these studies may have limitations, they represent both microarray and RNAseq data and the manuscript under review relies heavily on the assumption that prod1 is expressed in a similar manner in axolotls to newts. This warrants the request for evidence to show when and where prod1 and tig1 are expressed in the axolotl beyond the scRNAseq dataset and qPCR. Considering commercial FISH options are available that are somewhat quantitative and this is such a high impact paper and important question in the field, it seems reasonable to ask to show expression patterns in fibroblasts in either regenerating limbs similar to those used for the scRNAseq or proximal versus distal blastemas. This request does not discount the functional data presented, but the request seems reasonable considering the technologies are available and there are so many unanswered questions in terms of mRNA and protein expression related to prod1 (and tig1 by association).

Other, more minor critiques:

In figure 1e, Tig1 is upregulated after RA treatment without amputation. It is known that cells are not reprogrammed in uninjured limbs. It is unclear why Tig1 would change expression if it is the upstream regulator of PD patterning. Does the gene expression go back to baseline shortly after removing RA from the system?

In figure 1F, is it possible to replot using the raw data. The figure is distorted, I think because it is a screen shot off of axolotl-omics.

I am missing the utility of the Meandros image analysis tool. It doesn't seem like a neural network approach is necessary to find the outline of the limb...or I must be missing some explanation on this. It seems that a neural network approach would be better utilized for identifying a positive cell over background rather than tracing the limb. As it reads now, it seems that the most sophisticated method was used for the easiest part of ROI detection and the simpler method (thresholding) was used to identify the most difficult part of image quantification (identification of positive cells).

Figure 2 shows the limbs are shorter overall instead of having significant missing elements (or can this be quantified?). This would suggest that Tig1 is associated with cell proliferation more than identity of limb segments. I know this is mentioned, but maybe discuss a bit more. What is the phenotype for overexpression of prod1?

Page 18, line 6: I'm confused on the term "flight" here. What is this referring to?

Page 18, line 9: "bright filed" should be changed to "brightfield"

Figure 5a: "pde" should be "dpe"

Figure 5c: Need higher resolution for the samples.

Supplemental figure 4: The figure says anti-EdU. I'm guessing this needs to be changed to just EdU.

Best regards,

James Monaghan
Northeastern University
Department of Biology
Institute for Chemical Imaging of Living Systems
Boston, MA 02115

Reviewer #3:

Remarks to the Author:

BACKGROUND/CONTEXT:

This work aims to address a major outstanding question in regenerative biology about how information is encoded to ensure the regenerate structure has the proper anatomical axes and downstream features as the original structure. With respect to limbs, much more is known about how axes are specified and how cells execute broad-scale axis specification in terms of tissue placement. The same is not true for regeneration. The extent to which cells "remember" where they are in an uninjured limb and how this information might be imprinted on cells as they respond to an amputation injury, reorganize, and potentially re-set axes is largely unknown. In salamander limb regeneration, some published information has emerged showing that some aspects of positional memory are shared with limb development in organisms such as chick and mouse. This includes, for instance, that diminution of Sonic hedgehog signaling across the board leads to abolishment of the anterior-posterior axis in the autopod, while introducing an ectopic source of Shh along the anterior margin of the blastema mimics the mirror-image digit duplications observed when a similar experiment is performed in chick embryos (Roy et al., 1999, 2000; Riddle et al., 1993). There is also evidence, posted on BioRxiv (Takeuchi et al.), in newts that perturbing the Hox13 genes using a CRISPR-based gene editing strategy results in defects in limb development and regeneration predicted from orthologous loss-of-function experiments in mouse (Fromtenal-Ramain et al., 1996), specifically in patterning the proximo-distal axis. A central role for retinoic acid (RA) signaling in regenerating salamander limbs was first shown in 1982 (Maden) and several of these more genetic studies have defined some of the pathways responsive to RA treatment (such as Mercader et al., 2005; Voss et al., 2017; Nguyen et al. 2017; Monaghan & Maden; 2012; and others), but much remains to be elucidated about how this system operates to specify P/D in a regenerating limb and how this information is translated into patterning outcomes on a large scale and on a cellular scale.

Despite possible conservation of mechanisms between development and regeneration, differences are also possible, and these differences are important to elucidate and understand, especially because some might relate to the ability of salamanders to fully regenerate limbs while other tetrapods cannot. One such possible example is the work of Jeremy Brockes's lab in which they identified Prod1 as a GPI-linked (in newts, though in axolotls it is unanchored) cell-surface molecule that binds the Anterior gradient (nAG) ligand, provided evidence it may signal through EGFR, and demonstrated that Prod1 transcripts were enriched proximally along the P/D limb axis. They also showed that application of purported function-blocking antibodies against Prod1 can abolish the typical behavior of explanted blastema derived from proximal amputations whereby they engulf blastema derived from distal amputation *in vitro*. They subsequently showed that expression of nAG can rescue the effects of denervation and restore limb regeneration. This connection between proximo-distal patterning and

nerve requirement in regeneration remains murky. Nonetheless, the Prod1/nAG story has stood as sort of a prime example of the identification of a couple molecules implicated somehow in the P/D axis of regenerating limbs but for which we still have an incomplete understanding of how they are really operating.

SYNOPSIS OF THIS MANUSCRIPT:

Murky is a word we might use to describe much of our understanding, at this point, of how cells keep track of place in limbs from highly regenerative species and what happens to these memories during regeneration. Anything learned about this process, should it be robustly demonstrated with proper experimentation, stands to inform the broader picture of limb regeneration in general and, thus, regenerative medicine efforts. This manuscript hopes to make some of our understanding of these processes, specifically with respect to the proximo-distal axis, less murky. It is therefore a worthwhile pursuit and this research could lead to important contributions to this big and important question. Overall, the authors have made a genuine effort to bring a novel gene/molecule to our attention and have done some characterization to show that it could be a central player in how the proximo-distal axis is maintained and then patterned in regenerating limbs. However, this report needs considerable additional experimentation if the conclusions made are to be deemed defensible. The argument that Tig1 expression is graded along the P/D axis needs more support, and having a live reporter animal with a fluorophore knocked into the endogenous promoter (or some similar experiment) might provide that. A major limitation is that no genuine loss-of-function data is reported for Tig1 in an in vivo setting; the only approximation is this in vitro/explant blastema engulfment assay, which itself needs better controls, quantification, and exploration of effects at a cellular/molecular level. Additionally, some of the effects seen with the experimentation, even if appropriate controls and statistics are provided in revised experimentation, provide weak support that this Tig1 molecule is indeed a master regulator of this process rather than a downstream effector—one among many—of another central command pathway, perhaps the retinoid acid signaling pathway. The authors have configured their paper as a quest to fulfill a definition of a mediator of proximal identity to “exhibit a graded expression along the PD axis, be present at the cell surface, and be upregulated by RA.” This definition seems dated in an age where more robust molecular genetic techniques are now available in this model organism; rather than simply being descriptive of a gene/protein’s expression pattern and showing it’s induced by RA, defining the function of such a factor in the process of P/D specification and execution of patterning seems a better goal in the modern age, and it is achievable. Overall, this report presents preliminary findings, most of which need deeper investigation.

SPECIFIC CONCERNS:

- The identity of Tig1 seems to solely rely on the sequence alignment between the putative axolotl Tig1 and the human Tig1 versus other proteins. Given the similarities of Tig1 with CD38, the authors should strengthen these claims by performing a phylogenetic tree analysis of Tig1 and CD38 genes to verify the proper annotation. Relatedly, it appears that there are indications from the literature (Bryant et al., 2017) that CD38 itself may have a role in P/D axis in regenerating axolotl limbs, and so the authors should include this in their discussion.
- In other previously-published axolotl gene expression datasets (Campbell et al., Knapp et al., Stewart et al., Voss et al., Bryant et al., Gerber et al., Leigh et al., etc.) what does Tig1 transcript expression look like in those studies? Is there any more information about the other tissues that may express this gene other than fibroblasts? Any indication from these other studies that Tig1 has a graded expression? Can they find the gene in some of the studies that have centered explicitly on the effects of RA (Voss et al., Nguyen et al.)?
- Figure 1a: Can the authors comment on why some of the cells might be highly expressing both Tig1

and *Hoxa13* transcripts? The paper's model proposes their expression should be anti-correlated, but there appear to be several example cells on these graphs that show the opposite relationship. Is it possible that the principal component used to order the cells is not the best choice?

- Figure 1c: The statistics performed are not appropriate. Here a one-way ANOVA with post-hoc tests should be used. The expression appears to be decreasing but the Y-axis scale is misleading, this should follow what was used in Figure 1e. These data are used to validate the graded expression of the gene and they are weak. In situ hybridization should be used in addition to what is presented. It is also unclear as currently presented in Figure 1c whether there was first a purification or enrichment for connective tissues or cells before generating the cDNA for the three sample sites. If this were performed, then perhaps the authors would see a more robust and convincing proximal enrichment of the transcript. The published *prrx1*:GFP transgenic axolotls could be used to purify those fibroblasts and then used to generate cDNA from various P/D points along the limb axis. Furthermore, without any experiments to determine if other cells outside of connective tissue express *Tig1*, readers have no idea if the weak signal reflects the fact that this transcript is expressed in other cell types (and doing something else there?) and the contribution to total expression the authors care about is just swamped out. Purification followed by qPCR as well as a quantitative in situ method that includes full-length limb sections as well as magnified sections from several places highlighting connective tissues along P/D axis should be performed.

- For Figure 1c and 1e, individual data points should be plotted rather than reporting the data as bar or line graphs without this information available to the reader.

- Figure 1f: The microarray data suggests the peak of *Tig1* transcript expression is likely to be around 3 days post-amputation. Why, then, in Supplementary Figure 5 for their own qPCR analysis, did they start the regeneration time course samples at 9 days post-amputation? There's a lot of information missing in Supplementary Figure 5 about the expression of the gene in focus in the 9 days between amputation and first sampling.

- Overall, Figure 1 is so much less refined than is needed if there is to be a robust argument made that this factor is a strong candidate for conferring positional information regarding P/D axis in connective tissue cells and that it is responsive (i.e., transcription goes up) to retinoic acid treatment.

- Figure 2 and discussion of experiments therein: the effects observed following expression of *Tig1* in blastemas using baculovirus do not necessarily support a role for *Tig1* in patterning the P/D axis of the limb. Skeletal defects are presented, but there is no convincing argument these are not caused by another mechanism. The simplest prediction would have been that overexpression of a proximalizing factor, which *Tig1* is proposed to be, would lead to lengthening of the proximal elements or perhaps reduction of more distal elements at the expense of proximal. However, they observe malformations that cannot simply be explained by a proximalization (such as "malformations of carpals and metacarpals" which are binned separately from shortening or truncations, yet no example photos are shown). The graph gives the reader little solace that the effects are specific to P/D since they have not broken the information down any more than "limb length." This data would be stronger if they had simply measured all the humerus elements, all the radius/ulna elements, all the carpal elements, and all the digits from all the specimens and reported the data more granularly. Again, the individual data points should be presented rather than bar graphs, and so there is more info than the reader gets to see. Additionally, there is zero consideration given here to whether infection events were uniform across the samples. Disentangling this information and presenting it in a more holistic manner would allow the reader to decide if more extreme phenotypes are attributable to "better" infections, and the authors may even stand to improve the believability of their assertions if they were to simply re-analyze the data they've already collected or do the experiments again in a way that this information can indeed be reported.

- The authors generated a plasmid encoding a mutant *Tig1* to use as a loss-of-function control, but it

appears that it does not function that way. The use of the mutant does not help further the claims made in the paper. For instance, in the displacement assay presented in Figure 3c, the results show that the wild-type Tig1 has a very small effect to the proximal displacement of blastema cells. The mutant Tig1 has a bigger effect which cannot be explained. Another way to look at this is to simply say that the mutant Tig1 (P155A) has a neomorphic activity with respect to normal Tig1 function and that it could be doing something entirely unrelated to what the wild-type protein normally does. How do we even know this is a loss-of-function mutation rather than a neomorphic mutation? Are they hoping it acts as a dominant-negative? Where's the hard data to support that notion, if so? Another issue here is that the reader is left with approximating whether the differences in percentages among the different samples are statistically significant only by visual observation let alone assessing whether the difference between the percentages on the lower arm for RFP and TIG1wt have a biological meaning and support the authors claims that TIG1 induces distal-to-proximal displacement of blastema cells. The use of the TIG1 mutant, in this case, may be to distract the reader from the main issue and that is that the function and the role of wild-type TIG1 and its importance in normal P/D patterning, whereas the "loss-of-function" mutant should show results similar to that of RFP and not the opposite. Overall, looking at the fluorescent information at Figure 3d, Supplementary Figures 6 & 7, or the summarized data in Figure 3b, the effect of Tig1 appears minimal. The authors should consider strengthening these data with some statistical analysis otherwise it is hard to extract any actual biological meaning.

- The data in Figure 4 should be parsed with more granularity. Choosing to bin the results as either "engulfment" or "no engulfment" is misleading (because we can see intermingling at the cellular level even in the photos presented for what they claim is the most extreme case of non-engulfment, application of the Tig1 antibody). The authors should assay this result with more granularity. What percentage of RFP+ cells end up in the distally-derived tissue in each experimental scenario? Also, what is going on with the vastly larger tissue sizes in the examples treated with the antibody to Tig1? Were these blastema bigger from the beginning, or are they larger by virtue of more proliferation following explant? Or are these images not representative of the whole cohort examined? As presented, the first thing that could be concluded, if this antibody truly blocks Tig1 function, is that Tig1 might be a candidate tumor suppressor. Another issue is specificity of the reagent. Is aRFP the correct control for the engulfment assay? An antibody that binds to a cell surface protein and is considered not be functioning during this process should be used as control instead in order to test whether just having antibodies on cells affect their migration.
- The antibodies used have not been adequately characterized for readers to believe they are specific. Spec sheets from the company commissioned to generate the antibodies were provided, but these are irrelevant to the application of these antibodies. Evidence the antibodies are specific on western blot from protein extracted from axolotl tissues, on immunohistochemistry of axolotl tissues, and on the behavior of cells in vitro is lacking. The authors have made no attempt to perform peptide-blocking experiments that could help address these concerns; for example, any attempt to out-compete the supposed effect of the antibody in the engulfment assay with the peptide that served as the antigen.
- Again, in figure 5, there needs to be a better justification for believing the P155A mutation is effectually a loss-of-function or dominant-negative allele of Tig1. See previous comment about this. There are a variety of other interpretations for the data they present besides an overt role in orchestrating proximal-distal patterning in regeneration. The HoxA13:mCherry transgenic axolotl line is a nice addition to the paper and the field. Are the transgenics used non-mosaic? It was hard to tell from the methods if this is the case or not. Certainly, F0/mosaic animals would not be appropriate for this kind of experiment because we need to believe that all cells have the knock-in allele and that only some are turning the fluorophore on by virtue of them also expressing HoxA13 rather than the more trivial explanation that they were simply not edited at either HoxA13 allele.
- Tig1 overexpression does not affect meis1 expression as shown from the qPCR data in Figure 6 and had variable expression in the RNAseq dataset (no standard deviation bars are presented in Figure 5).

Figure 6 shows no effect on *mesi1*, and minimal effect on *prod1*, so the author's claims are not supported. The statistics for Figure 6 are not appropriate. One-way ANOVA with post hoc tests should be used instead if the variances among the samples are equal and equivalent non-parametric tests should be used if they are not (which in many of them seem to be the case based on the standard deviation presented here).

- The authors should mention that the statistics presented for the RNAseq expression data are q-values and not p-values as indicated in the legend (Figure 5).
- An integral issue with the mCherry+ (HOXA13+) sorting of the cells and RNAseq is that the *Tig1* overexpression downregulates the HOXA13 expression which controls the mCherry. In addition, *Tig1*+ cells will assume more proximal positions in the blastema (according to the authors data and claims), thus the "reprogramming" of their transcriptomic profiles may be not due to the direct effect of the *Tig1* overexpression but rather the effects of other PD-related molecules including retinoic acid.

We thank all Reviewers for their helpful comments and suggestions, which helped improving our manuscript. In the following response, we address (in blue) the specific comments of each Reviewer.

REVIEWER COMMENTS

Reviewer #1 (Remarks to the Author):

This is a nicely written paper that describes a new protein candidate for proximal positional identity within the axolotl salamander arm. Previously, Prod1 was identified as a proximal positional identity protein in the newt and axolotl. Data are presented in this paper to argue that Tig1 also satisfies criteria of a proximal positional identity protein. Tig1 is clearly an interesting molecule and appears to affect aspects of cellular identity and limb regeneration given the experiments that were performed, and importantly results from this study further implicate a relationship between cell proliferation and proximalisation. However, the connection between Prod1 and Tig1 in determining proximal identity remains unknown and some issues need to be clarified/addressed in the manuscript to better understand the significance of the work and whether or not it will move thinking beyond what has already been established from studies of Prod1.

We thank the reviewer for acknowledging the importance of our work in characterizing TIG1 as a novel candidate of PD identity determination. Beyond the classical functional experiments in the field to test for PD fate conversion, of which we provide extended evidence in this revised version, we demonstrate the capacity of one molecule, TIG1, to induce global changes at the transcriptomic level towards proximal identity. Importantly, this is the first example of a cell surface molecule able to induce global reprogramming of proximo-distal identity, thus constituting a significant advance beyond what is currently known in the field.

This important notion is now also supported by an additional in-depth bioinformatic analysis, combining our dataset with the recently-published Lin et al 2021 (including Prod1, which is found in connective tissue and proximally localised). We show that the transcriptome of TIG1-transformed distal blastemal cells shift towards a naturally-occurring proximal region-transcriptional signature of the early-stage blastema, highlighting a likely central role of TIG1 in PD fate determination during regeneration. Further, our findings on the impact of Tig1 on the proximo-distal transcriptomic landscape include the modulation of HoxA13 expression and its downstream network, PROD1 and Tig1 itself, hinting at a higher level at the pathway hierarchy and thus providing a solid basis for further research explorations. We acknowledge these will be required for elucidating the exact connection between Tig1 and Prod1 in PD identity, yet feel that this is beyond the scope of this manuscript.

We also feel that the extensive body of work presented in this revised version, which identifies and establishes Tig1 as novel determinant of proximal identity, and the only cell surface determinant known to reprogramme a cell's PD identity, constitutes a major conceptual advance towards understanding how positional identity is encoded. In this revised version, we have addressed and clarified the issues raised by all reviewers in order to improve the understanding of our findings and their significance.

Comments/Concerns

P2, Line 34. Does mature limb mean intact limb? This is a bit confusing to me and perhaps others that are not in the limb field.

In this manuscript 'mature limb' and 'intact limb' are terms used to refer to limbs from juvenile or adult axolotls that have not been manipulated. The nomenclature was standardized to 'mature limb' to avoid confusion.

It should probably be noted here that previous studies have not found a graded expression pattern of Meis expression in the mature/intact limb. Doesn't this contradict the principles that are proposed for a PD molecule and coordinate system?

The activity of Meis transcription factors has been previously shown to be regulated at transcriptional, post-transcriptional and post-translational levels. For instance, interaction between Meis and Pbx proteins is essential for Meis nuclear translocation and subsequent gene expression. Despite the absence of pronounced differences in Meis expression along the PD axis of the intact salamander limb, a higher activation of MEIS in the proximal blastema vs. distal blastema has been reported (Mercader, 2005; Shaik et al, 2011). This strongly suggests that Meis is regulated beyond the transcriptional level to induce differential activation in this context, which is in general agreement with the predictions of the existence of a PD axis gradual activation during limb regeneration.

In addition, we acknowledge that said “set of principles” does not represent a universal definition of a PD determinant but a logic that was successfully applied for identifying Prod1. This is also reflected in the revised manuscript.

P2, Line 35. Need to make it clear here that Prod1 with a GPI anchor was originally identified in the newt. Without this qualification it is confusing when the axolotl GPI-anchorless version is introduced.

We agree with the Reviewer and have modified the text to ‘This led to the identification of the GPI-anchored cell-surface molecule Prod1 in the newt, which fulfils all said criteria....’

P3, Line 22. This wording seems a bit strong.... “that accurately modelled changes”. A static principle component/variable calculated for a single sample/time point cannot model changes, whether they be accurate or inaccurate.

We agree with the Reviewer and have modified the text accordingly (P.3, Line 24)

P3, Line 24. Please state the total number of cells that were positive (relative to a threshold) for Hoxa13, Meis1, and Tig1. It seems that the resolution PC3 provides is minor (how much variance explained) and mostly associated with low Hoxa13 expression.

In this dataset we found that 90 out of 163 cells express Tig1, 43 out of 163 express HoxA13 and 61 out of 163 express Meis2; In all three cases the threshold is 1 or more transcripts per million. The percentage of variance for PC3 is 1.2% and this is now stated in the corresponding legend (New Suppl Fig 1). Of note, the POV for each PC in the Gerber dataset is as follows:

Figure 1 – Percentage of variance associated to each principal component analysed from the Gerber et al 2018 dataset .

P3, Line 24. The Figure 1 legend says that Prod1 expression is not available in the single cell dataset. Might Prod1 align to “AC_02200036472.1 in the transcriptome that was used to annotate the single cell data and if so, are there expression data for this transcript model?

If not, and Prod1 is expressed by blastema/connective tissue cells, how was it missed in the single cell study that enriched for connective tissue cells? It seems from other single cell data that Prod1 might be expressed highly in immune cells and less so in connective tissue cells. Might Prod1

expression associate with immune cells and not connective tissue cells or perhaps an immunological cell phenotype?

The contig (AC_02200036472.1) is indeed Prod1, however in the transcriptome version used in Gerber et al. AC_02200036472.1 was unannotated as the contigs were annotated with the best human match. Being unannotated, this contig would have been excluded from the analysis. To address this issue, we have analysed the newer dataset available from Lin et al 2021. Notably, we find PROD1⁺ cells located in connective tissue cells, in regions associated to clusters enriched in proximal markers MEIS 1 and 2 (new Fig. 1) and closely associated to Meis on a modelled PD axis (PC1), also defined by the same known markers (new Fig. 1b). This shows that PROD1 is a connective tissue factor with a strong association to proximal identity within the blastema.

Prod1 is a very lowly expressed gene which may explain its absence from some datasets, particularly in single cell ones where low expressed genes are often not detected. After careful examination, we weren't able to identify PROD1 in the Leigh et al 2018 dataset either (which includes immune cell population datasets). To our knowledge, other than connective tissue cells, PROD1 expression has only been associated to Schwann cells (Kumar et al 2007).

It would seem important to show *in situ* data for Prod1 and Tig1 in the manuscript as the proposed PD coordinate system assumes both molecules contribute in some unresolved way.

We agree that the spatial pattern of TIG1 and Prod1 endogenous expression important to our study. However, Prod1 molecule has been studied by others in more detail and *in situ* hybridization published (DaSilva et al 2002 and Kumar et al 2007), showing an association of PROD1 with the blastema mesenchyme during salamander limb regeneration, overall consistent with the notion that PROD1 is essentially a connective tissue factor. This is indeed supported by our new analysis of the Lin et al dataset, which indicates Prod1 is expressed in axolotl connective tissue cells enriched in proximal markers (**new Fig. 1**)

We now provide *in situ* hybridization for TIG1 in the regenerating limb. We show that upon upper arm amputation, TIG1 is strongly upregulated from 1 to 7 days post amputation, after which the expression gradually decreases and is seldomly detected at 14 dpa. The expression is evident in the proximal blastema and adjacent stump tissues. The temporal pattern observed is corroborated by previous microarray analysis from datasets published in Knapp et al 2013 (**new Fig. 2d**), and is in general agreement with the data published in Gerber et al 2018 in Fig3-j. Complementary, our RT-PCR analysis shows that TIG1 expression levels remain low during late stages of regeneration, which is also observed in the Knapp et al 2013 dataset. We also provide data for lower arm amputation. Here Tig1 expression follows the same general pattern (**new Suppl Fig. 8**).

In addition, we confirmed that TIG1 is mainly present in connective tissue cells, with over 70% of the ISH-labelled TIG1⁺ co-staining with PRRX1 antibody in 5- and 7-day blastema samples (including adjacent stump). This data was added as **new Suppl Fig 9**.

Despite that our RT-PCR detects a graded expression in the PD axis of the mature limb, we were not able to see significant expression of TIG1 in the mature limb samples using ISH, likely due to the lack of sensitivity of the technique. However, we found a similar differential expression pattern of TIG1 in the newt *Notophthalmus viridiscens* mature limb through RT-PCR, as the **new Suppl Fig 2** shows. Thus, Tig1 expression is enriched in proximal elements of the mature limb in both axolotls and newts.

We have attempted to label TIG1 and PROD1 proteins with specific antibodies, but we were unable to observe reliable signal in either case.

P3, Line 46. It is known that aneuploid AmAL1 cells express Prod1 and Meis1/2. Is this true for NvA1 cells?

NvA1 cells do not express Prod1, as shown by Da Silva et al 2002. According to our own data, Meis1/2 expression is negligible when compared to AL-1.

P4, Line 24. Not clear the data support the idea of strong Tig1 downregulation when there is unequal sampling of early vs late timepoints. With more late samples one might conclude there is a gradual decrease in transcripts.

We agree with the reviewer, the wording was probably confusing. In **Suppl Fig 7**, RT-PCR data covers the quantification of TIG1 expression from 9 dpa on, showing that transcripts gradually

decrease and remain at very a low level during the subsequent stages of regeneration. In addition, the RNAseq data on Fig3-j in Gerber et al 2018, also reports a gradual decrease of TIG1 (here labelled as 'RARRES1') transcripts from 8 dpa until 18dpa, which corroborates our findings.

P4, Line 37. It's not clear that the following conclusions - Tig1 establishes PD patterning and Tig1 is strongly implicated in determination of proximal identity – follow from the misexpression results. The data in this paragraph show a disruption of limb regeneration, which could trace to regulation of cell proliferation, which was shown in the preceding paragraph, or disruption of some other biological process.

We thank the reviewer for this comment. We agree that in the previous version of the manuscript, distinguishing a role of Tig1 in PD patterning versus overall cell cycle regulation with certainty was difficult. This led us to expand the experiment and its analysis, and we found clear reductions in limb lengths and malformations specific to the distal limb elements, while the proximal ones remained unaffected (see **new Fig 3** and associated text). Should the phenotype result solely from an impact on cell cycle regulation, or a non-PD related biological process, one would expect a general effect evenly distributed along the segments. Yet this is not the case. Further, the resulting phenotype is similar to that observed from overexpressing Prod1. Together, this new dataset strongly corroborates the role of Tig1 in the regulation of PD patterning of the regenerating limb.

P5, Line 19. Why is there such a big difference in how Prod1 vs Tig1 expression displaces cells? The mechanisms of action of these molecules are largely unknown. While it is possible that these molecules may act within the same network to promote proximalisation of blastema cells, as has been suggested by our RNAseq and RT-PCR data, their distinct phenotypes raise the possibility that overexpression of each factor may elicit different molecular outcomes in blastema cells, which are still to be determined. Alternatively, and as noted in the main text, it cannot be excluded that a dose-dependent effect underlies the differences observed.

Could Prod1+ cells include non-blastema migratory cells?

We think it is unlikely that the PROD1-transfected cells that proximalise correspond to non-blastema cells. Even in the samples in which proximalisation was very pronounced (please see previous comment), these cells remain relatively close to each other, a phenotype not typically seen in e.g. limb immune cells.

I don't really see an additive effect of Prod1 and Tig1, if anything, the combination of the two seems to reduce Prod1's overall effect. Perhaps this should be quantified to strengthen the argument?

We thank for the comment and following the reviewer's request, we performed new statistical analysis comparing the fluorescent signal between two conditions along the PD axis. The plots in new Suppl Fig 12 represent the significance values found along each region. With this analysis, we indeed observe a significant difference between the Prod1 and Prod1+Tig1 conditions, with Prod1-overexpression corresponding to a higher extent of translocation towards proximal regions.

We therefore altered the main text to reflect the new statistical analysis and eliminate the former assertion that Prod1+Tig1 combination show an additive effect, to now consider that instead a differential effect is present regarding the overexpression of Tig1 and Prod1 alone or in combination.

Nevertheless, we would like to note that this phenotype was more variable within the PROD1 condition than among the combination of TIG1 and PROD1, which, as observed in **Supp Fig. 13b**, manifested as more samples exhibiting translocation to lower arm and upper arm regions in the Prod1+Tig1 combination than with Prod1 alone, suggesting a more consistent effect is achieved in the former case.

We further note that, in light of the new in vitro analysis where we found that TIG1-overexpression hampers cell motility (included as new Supplementary Fig. 15 and discussed in a comment above), it is possible that a reduction in the translocation extent of the Prod1+Tig1 combination reflect a dominant effect resultant from the decreased the ability of Tig1-expressing cells to translocate, which can be independent of its reprogramming ability, seen in our RNAseq dataset.

P5, Line 25. These cell proliferation data do not provide much further resolution. Other important processes – migration and cell adhesion – are mentioned to be important but no data are provided.

We acknowledge this caveat. Therefore, we performed additional experiments to explore the hypothesis that migration or adhesion processes might underlie important aspects of TIG-induced phenotype. In particular, we established and performed a ‘wound healing/scratch assay’ with AL1 cells, and observed that TIG1-transfected cells display less efficient gap closure than control transfected cells and this effect is rescued using anti-axolotl TIG1 antibodies. The negative impact on invasiveness could be explained by alterations in cell adhesion properties. In this regard, we note that Tig1 overexpression modulates the expression of adhesion factors (Fig. 6h) some of which are proximally-enriched in our analysis (Supplementary Fig 1c), supporting this hypothesis. We have included these observations in the **new Suppl Fig 15** and commented on the text.

P6&7. The gene expression data should be made available as a supplementary file showing p values and fold change values. What criteria specifically were used to identify the top 2000 genes?

The caption and Figure 6 legend should probably indicate that Tg1 overexpression affects the expression of the indicated genes, not controls their expression.

We have added the requested data as **Supplementary data 7**. Also for the new analysis in Fig 8, we have added separate lists for the up or downregulated genes used (**Suppl data 9 and 10** respectively). Criteria used for top 2000 genes: the standard deviation across samples was calculated for all genes. These were then ranked by decreasing standard deviation and the top 2000 selected for the K-means clustering.

We have revised the legend of Figure 6 (now the **new Fig 7**).

P8, Line 42. The proposition that Tig1 is a mediator of RA-dependent proximalisation could be investigated further by comparing transcriptional changes mediated by Tig1 over-expression to those mediated by RA administration.

As suggested, we compared the RA treatment-related transcriptional changes dataset from Nguyen et al 2017 (table S1, cutoff of at least 1.5x down or upregulated, in a DMSO vs. RA comparison) to our TIG1-induced transcriptional changes at three days post electroporation. Notably, we found MEIS1 among the common significantly upregulated genes, and HoxA13, Lhx2 and SPRY1 among the common significantly downregulating genes, revealing an overlap regarding important PD marker genes. Thus, there is a concordance of effect at the gene expression level, as can be seen in the common up and downregulated charts. However, we would like to stress that a) the experimental design differ significantly between the two studies; b) RA is a molecule which not only regulates Tig1 expression but that of many other genes beyond a proximalisation context, and c) in Nguyen et al 2017 the whole blastema tissue was analysed, whilst in our experiment, TIG1 overexpression was induced very restrictively to distal blastema tissue, and only HoxA13+ distal cells were isolated and analysed.

Up in RA/ up in Tig1 (41 genes)	A4GNT	CETP	LPIN2	SLC4A4
	ABI1	CLN3	MEIS1	SLK
	ACAN	CYP2C8	NLRP12	SMPDL3B
	ADAM9	DHRS3	NR3C1	SOAT1
	AP2B1	DPP4	ORF2P	SWAP70
	B3GNT5	FABP2	PDIA5	TM6SF2
	B4GALT3	FAM102B	PLXNB2	TMX3
	BHLHE40	KRT15	PTBP3	TNFRSF1B
	C1GALT1	KRT8	SAMD9L	UGT2A1
	CDC42SE2	LGALS9	SDR16C5	VIL1
	CES2			

Down in RA/ down in Tig1 (27 genes)	AQP3	PMCH
	ARHGEF3	PRSS23
	BMPR1B	SLITRK6
	CDO1	SNTB1
	CRLF1	SOSTDC1
	DNER	SOX8
	EBF3	SPRY1
	HK2	SRSF1
	HoxA13	TFPI2
	LGR4	TMEFF2
	LHX2	UAP1
	LRRN1	UAP1
	LYPD6	ZIC5
	MSX2	

Figure 2 – Comparison of the transcription profile between RA-treated blastemas (Nguyen et al 2017 dataset) and TIG1-overexpression in distal blastemas (current manuscript dataset) shows that these treatments share a number of expression changes of PD markers,

notably *Meis1* (downregulated), *Hoxa13*, *Lhx2* and *Spry1* (upregulated), highlighted in yellow. Commonly upregulated and downregulated gene lists are represented in left and right, respectively.

Reviewer #2 (Remarks to the Author):

This is an exciting manuscript by Oliveira et al presenting evidence that the molecule *Tig1* is important in determining the positional information of fibroblasts during salamander limb regeneration. How positional information is stored and utilized in salamander limbs in order to regenerate the appropriate proximodistal structures is a longstanding and important question in the field.

Noteworthy results of the manuscript include the identification of a new molecule that can change the positional information of limb cells in a regenerating salamander limb. Few other molecules (*prod1* and *meis1* and 2) have been shown to have similar properties so the identification of a new molecule is noteworthy. As a side note, it is unclear how many genes may generate similar phenotypes. The work will have considerable impact in the field of limb regeneration and I assume that many others studying limb development will also take notice. Discovery of the previous molecule, *prod1* (or C59), has had a lasting impact on the field of regeneration biology, where people have studied its role in a number of species including lizards and zebrafish appendage regeneration. How positional information is maintained and established during limb regeneration is still a very open question and is of critical importance for our understanding of regeneration. Although the experiments presented here are classic in the field, the novelty is in the molecule identified.

The experiments do support the conclusions although the experiments are slightly limited, likely due to limitations with the current methods used in the animal model and the significant amount of work it takes to generate genetically modified axolotls. The study meets the standard in the field (axolotl), although the phenotypes presented are less severe than expected, which may be due to the limitations of electroporation and the new techniques of baculovirus rather than the function of the gene of interest. The engulfment assay phenotypes are also somewhat limited compared to sorting assays that are performed in chicks and mice limb bud experiments. These are more limitations of the current methodologies used in axolotls rather than the approaches used in the study. Overall, I applaud the use of *in vivo*, *ex vivo*, and *in vitro* functional approaches to address the problem. The authors used the appropriate techniques available in the animal model to address the question.

Although the gain of function experiments are overall convincing, I'm missing the spatial expression profiles of *prod1* and *tig1* in the limb. Is *Tig1* expressed in all connective tissue fibroblasts? RNAseq isn't enough for me considering only *Prrx+* cells were studied. Also, the expression pattern of *prod1* hasn't really been shown with modern FISH techniques. I think this is a general weakness of the manuscript.

We thank the reviewer for the appreciation of our work and the significance of characterizing *TIG1* in the field of regeneration. We agree with the comments and now provide *in situ* hybridization profiles for *TIG1* in the regenerating limb. We show that *TIG1* is strongly upregulated during the first 7 days after upper arm amputation, after which expression decreases until 14 dpa (**new Fig 2e**). The expression is seen in the proximal blastema and adjacent stump tissues. Lower arm amputation follows the same pattern (**new Suppl Fig 8**). Despite that our RT-PCR detects a graded expression in the PD axis of the mature limb, we were not able to detect expression of *TIG1* in the mature limb by *in situ* technique, likely due to lack of sensitivity of the technique. This is now mentioned in the manuscript.

In the revised version, we additionally show that *TIG1* is mainly expressed in connective tissue fibroblasts (70%), as confirmed by our ISH-labelled *TIG1*⁺ co-stained with a *PRRX1* antibody in a 5- and 7-day blastema, including adjacent stump (**new Suppl Fig 9**).

In situ hybridization for *PROD1* has been published (DaSilva et al 2002 and Kumar et al 2007), showing an association of *PROD1* with the blastema mesenchyme during salamander limb regeneration, in a pattern consistent with connective tissue. Unfortunately, despite our efforts to

conduct ISH and IF for PROD1 in axolotl tissues, we were unable to obtain reliable staining. Nevertheless, our new bioinformatics analysis of the Lin et al, 2021 dataset offers new insights into axolotl Prod1, indicating that it is expressed in connective tissue cells enriched in proximal markers (**new Fig. 1**).

In relation to the above statement, I suggest modification in the writing, especially in the introduction, to discriminate between experiments performed in axolotls versus newts. The authors build the story as if it is all cohesive between species. Much of the early work on prod1 was done on newts, and much of the grafting work on axolotl. Considering the differences in prod1 localization (cell surface versus secreted), differences on muscle regeneration (dedifferentiation versus stem cell based), and differences in the impact of denervation (permanent inhibition versus temporary), it is becoming less convincing to extrapolate results in one species to the other.

This point is especially true in the following statement at the end of the introduction: "This raises the possibility that, in order to achieve a graded distribution along the PD axis and execute its functions, Prod1 interacts with additional, as yet unidentified components of the proximo-distal memory system."

Several studies contradict this statement in axolotls. Prod1 was higher in distal blastemas compared to proximal blastemas according to McCusker et al., 2015, prod1 did not respond to RA treatment in Nguyen et al., 2020 (I just analyzed the data on NCBI GEO database to doublecheck), and did not show up as differentially expressed between proximal and distal samples in Bryant et al 2017 or Voss et al 2018. Although each of these studies may have limitations, they represent both microarray and RNAseq data and the manuscript under review relies heavily on the assumption that prod1 is expressed in a similar manner in axolotls to newts.

We fully agree that comparison of regeneration between species at the cellular and molecular levels, must be done cautiously. We have made modifications in the manuscript to better discern what is known in newt vs. axolotl.

We are aware that studies have revealed disparity and inconsistencies regarding PROD1 expression. This is partially explained by its low expression in normal conditions as well as the lack of its annotation in some of the reported datasets (we checked this and were not able to find Prod1 in most datasets, thus it is not possible to draw conclusions on this molecule). As an example, Prod1 was not originally annotated as such in the Lin dataset (2021), thus we had to identify the transcript sequences corresponding to Prod1, annotate it as such and reanalyse the dataset. This is a common problem with salamander gene annotation. Following the Reviewer's concern, we then investigated this dataset further (**new Fig 1**) and found that Prod1 is expressed in connective tissue cells, in the proximal cluster (as defined by expression of other markers such as Meis1 and 2, also the cluster which contains Tig1+ cells). This suggests that Prod1 is associated with proximal fates in the axolotls, as originally proposed in newts, and is consistent with previous experiments performed on axolotl Prod1 which show it responds to RA (Shaikh et al, 2011), and it promotes proximal displacement (Echeverri and Tanaka 2005)

In the revised version we also expand our understanding of TIG1 expression in the newt (*Notophthalmus viridescens*). We found that in *N. viridescens*, similarly to the axolotl, TIG1 transcripts are more enriched in the proximal regions of the limb, as examined by RT-PCR (**new Suppl Fig 2**), demonstrating this important aspect is conserved between the two species. Furthermore, we analysed an unpublished scRNAseq data from whole mesenchymal blastema tissues of adult *N. viridescens* at 18 dpa (equivalent stage to 11dpa in 9cm axolotls), revealing that TIG1 expression is again associated with the extreme proximal end of the modelled PD axis (based on the expression profile of the markers HoxA13⁺ and Meis2⁺). We present the data here for the reviewers' consideration; however, the entire dataset will be subject of an upcoming publication (Simon lab, in preparation) and thus we are unable incorporate it in our revised manuscript.

Fig. 3 – Principal component analysis of scRNAseq from a *Notophthalmus viridescens* 18 day blastema. PC5 models the PD-axis, as evidenced by opposing the distribution of cells highly enriched in transcripts of the proximal marker *Meis2* and the distal marker *Hoxa13* (in red). *Tig1* is found among the modelled proximally-enriched genes, as in the axolotl case.

This warrants the request for evidence to show when and where *prod1* and *tig1* are expressed in the axolotl beyond the scRNAseq dataset and qPCR. Considering commercial FISH options are available that are somewhat quantitative and this is such a high impact paper and important question in the field, it seems reasonable to ask to show expression patterns in fibroblasts in either regenerating limbs similar to those used for the scRNAseq or proximal versus distal blastemas. This request does not discount the functional data presented, but the request seems reasonable considering the technologies are available and there are so many unanswered questions in terms of mRNA and protein expression related to *prod1* (and *tig1* by association). Please refer to last comment regarding the new ISH for TIG1 data added to the manuscript.

We fully acknowledge the gap of knowledge that exists in the field regarding PROD1, and the importance of understanding a potential connection between PROD1 and TIG1. Indeed, in the revised manuscript we providing evidence of Prod1 presence in connective tissue cells of proximal nature in the axolotl, as well as data suggesting its regulation by TIG1. We would also like to stress that we have focused our efforts on identifying a novel component of the PD determination system, TIG1, and in extensively characterising its impact on PD identity. We feel that further in-depth characterization of the PROD1 is beyond the scope of the present manuscript.

Other, more minor critiques:

In figure 1e, *Tig1* is upregulated after RA treatment without amputation. It is known that cells are not reprogrammed in uninjured limbs. It is unclear why *Tig1* would change expression if it is the upstream regulator of PD patterning. Does the gene expression go back to baseline shortly after removing RA from the system?

We think this is a fair point but we note that there is a very different environment between the blastema and the intact limb, and that unknown factors can be at play to promote strong stabilization of identity in the context of mature limb. In the later context, TIG1 expression might not be sufficient by itself to execute reprogramming in intact limb fibroblasts but, the conjunction of injury signals and the many molecular pathways activated during regeneration may create permissive environment for reprogramming to occur.

To investigate the expression dynamics of TIG1 in the mature limb upon RA treatment, intact animals received an intraperitoneal injection of freshly made stocks of 30 mg/ml all-trans retinoic

acid, at 100 µg/g body weight. Our results show an increase of TIG1 expression in the mature limb until the last timepoint analysed (9 days), see graph below. Some animals were amputated after injection, as a control for the activity of all-trans retinoic acid (images on right side exhibiting axis elongation and digit malformations).

However there is an important technical limitation which limits the conclusions that can be drawn from this experiment, namely the formation of crystals in the intraperitoneal area, which remain undissolved throughout the time points analysed. We note that these crystals act as a source of long-term release of RA, from which one would expect to influence TIG1 expression, explaining the progressive increase in TIG1 expression.

Figure 4 – All-trans-retinoic acid treatment via intraperitoneal injection resulted in continual increase of TIG1 transcripts, as measured by RT-PCR analysis (left). Morphological alterations highlighted by the arrows (right) correspond to the expected alterations induced by RA during limb regeneration, confirming the efficacy of the treatment.

In figure 1F, is it possible to replot using the raw data. The figure is distorted, I think because it is a screen shot off of axolotl-omics.

We agree, the new figure is now on **Fig 2d**, containing individual data for the 6 microarray probes analysed.

I am missing the utility of the Meandros image analysis tool. It doesn't seem like a neural network approach is necessary to find the outline of the limb...or I must be missing some explanation on this. It seems that a neural network approach would be better utilized for identifying a positive cell over background rather than tracing the limb. As it reads now, it seems that the most sophisticated method was used for the easiest part of ROI detection and the simpler method (thresholding) was used to identify the most difficult part of image quantification (identification of positive cells).

We thank the reviewer for the comment. The reviewer is right in that the neural network approach of the ROI-detection module could be also used to identify positive cells over the background. However, to quantify the spatial distribution of cells within the entire limb requires a spatial resolution in which the thresholding method is more than sufficient. Moreover, the ROI-detection module does save user time when compared with manual detection of the limb outline manually. Nevertheless, the most important advantage of Meandros is that it calculates the distribution of cells (or signals) projected along any curve within the tissue. In this particular problem, we wanted to determine the distribution of cells along the anatomical proximal-distal axis, which is characterized by its clear curvature. Thus, Meandros solves the problem of how to project the cellular information acquired from the whole tissue upon this non-Euclidean anatomical axis. On top of that, it allows the user to run statistical tests on the flight comparing different conditions (see the answer to Reviewer #3)

Figure 2 shows the limbs are shorter overall instead of having significant missing elements (or can this be quantified?). This would suggest that TIG1 is associated with cell proliferation more than identity of limb segments. I know this is mentioned, but maybe discuss a bit more. What is the phenotype for overexpression of prod1?

The phenotype for Prod1 overexpression (in axolotls) is published in Echeverri and Tanaka 2005 and closely resembles our observations for Tig1, namely missing and truncated digits, malformed metacarpals and shortening of distal skeletal elements.

We have now expanded our experimental dataset and analysis (**new Fig 3**), including new replicates, limb segment ratio quantifications and additional examples of the skeletal abnormalities found. Several elements from the distal-most regions were morphologically altered, including digit fusion, digit truncation and fusions and carpal malformations; occasionally radius and ulna distal epiphysis were elongated or the bones were fused. We provide more refined analysis of each segment length including the visualization of individual points. Briefly, analysis of the TIG1-transfected limb versus its contralateral control revealed that there is a significant reduction in size associated with TIG1 Treatment; Furthermore, there is a trend for size reduction in the lower arm and a significant downsizing of the hand. Conversely, the upper arm length is comparable to the control samples. The kernel density graphs now included in **new Fig 3d** in the manuscript allow to clearly visualize these shifts. Importantly, this phenotype suggests a specific effect on the regeneration of distal elements, beyond any general impact on cell proliferation (which one would expect to affect all segments equally). This is now further discussed on the text (P.5 Line12)

Page 18, line 6: I'm confused on the term "flight" here. What is this referring to?

Page 18, line 9: "bright filed" should be changed to "brightfield"

Figure 5a: "pde" should be "dpe"

Figure 5c: Need higher resolution for the samples.

Supplemental figure 4: The figure says anti-EdU. I'm guessing this needs to be changed to just EdU.

We thank the Reviewer for the comments. These changes have been implemented in the revised version.

Best regards,

James Monaghan
Northeastern University
Department of Biology
Institute for Chemical Imaging of Living Systems
Boston, MA 02115

Reviewer #3 (Remarks to the Author):

BACKGROUND/CONTEXT:

This work aims to address a major outstanding question in regenerative biology about how information is encoded to ensure the regenerate structure has the proper anatomical axes and downstream features as the original structure. With respect to limbs, much more is known about how axes are specified and how cells execute broad-scale axis specification in terms of tissue placement. The same is not true for regeneration. The extent to which cells "remember" where they are in an uninjured limb and how this information might be imprinted on cells as they respond to an amputation injury, reorganize, and potentially re-set axes is largely unknown. In salamander limb regeneration, some published information has emerged showing that some aspects of positional memory are shared with limb development in organisms such as chick and mouse. This includes, for instance, that diminution of Sonic hedgehog signaling across the board leads to abolishment of the anterior-posterior axis in the autopod, while introducing an ectopic source of Shh along the anterior margin of the blastema mimics the mirror-image digit duplications observed when a similar experiment is performed in chick embryos (Roy et al., 1999, 2000; Riddle et al., 1993). There is also evidence, posted on BioRxiv (Takeuchi et al.), in newts that perturbing the Hox13 genes using a CRISPR-based gene editing strategy results in defects in limb development and regeneration predicted from orthologous loss-of-function experiments in mouse (Fromental-Ramain et al., 1996), specifically in patterning the proximo-distal axis. A

central role for retinoic acid (RA) signaling in regenerating salamander limbs was first shown in 1982 (Maden) and several of these more genetic studies have defined some of the pathways responsive to RA treatment (such as Mercader et al., 2005; Voss et al., 2017; Nguyen et al. 2017; Monaghan & Maden; 2012; and others), but much remains to be elucidated about how this system operates to specify P/D in a regenerating limb and how this information is translated into patterning outcomes on a large scale and on a cellular scale.

Despite possible conservation of mechanisms between development and regeneration, differences are also possible, and these differences are important to elucidate and understand, especially because some might relate to the ability of salamanders to fully regenerate limbs while other tetrapods cannot. One such possible example is the work of Jeremy Brockes's lab in which they identified Prod1 as a GPI-linked (in newts, though in axolotls it is unanchored) cell-surface molecule that binds the Anterior gradient (nAG) ligand, provided evidence it may signal through EGFR, and demonstrated that Prod1 transcripts were enriched proximally along the P/D limb axis. They also showed that application of purported function-blocking antibodies against Prod1 can abolish the typical behavior of explanted blastema derived from proximal amputations whereby they engulf blastema derived from distal amputation *in vitro*. They subsequently showed that expression of nAG can rescue the effects of denervation and restore limb regeneration. This connection between proximo-distal patterning and nerve requirement in regeneration remains murky. Nonetheless, the Prod1/nAG story has stood as sort of a prime example of the identification of a couple molecules implicated somehow in the P/D axis of regenerating limbs but for which we still have an incomplete understanding of how they are really operating.

SYNOPSIS OF THIS MANUSCRIPT:

Murky is a word we might use to describe much of our understanding, at this point, of how cells keep track of place in limbs from highly regenerative species and what happens to these memories during regeneration. Anything learned about this process, should it be robustly demonstrated with proper experimentation, stands to inform the broader picture of limb regeneration in general and, thus, regenerative medicine efforts. This manuscript hopes to make some of our understanding of these processes, specifically with respect to the proximo-distal axis, less murky. It is therefore a worthwhile pursuit and this research could lead to important contributions to this big and important question. Overall, the authors have made a genuine effort to bring a novel gene/molecule to our attention and have done some characterization to show that it could be a central player in how the proximo-distal axis is maintained and then patterned in regenerating limbs. However, this report needs considerable additional experimentation if the conclusions made are to be deemed defensible. The argument that Tig1 expression is graded along the P/D axis needs more support, and having a live reporter animal with a fluorophore knocked into the endogenous promoter (or some similar experiment) might provide that. A major limitation is that no genuine loss-of-function data is reported for Tig1 in an *in vivo* setting; the only approximation is this *in vitro*/explant blastema engulfment assay, which itself needs better controls, quantification, and exploration of effects at a cellular/molecular level. Additionally, some of the effects seen with the experimentation, even if appropriate controls and statistics are provided in revised experimentation, provide weak support that this Tig1 molecule is indeed a master regulator of this process rather than a downstream effector—one among many—of another central command pathway, perhaps the retinoid acid signaling pathway. The authors have configured their paper as a quest to fulfill a definition of a mediator of proximal identity to “exhibit a graded expression along the PD axis, be present at the cell surface, and be upregulated by RA.” This definition seems dated in an age where more robust molecular genetic techniques are now available in this model organism; rather than simply being descriptive of a gene/protein's expression pattern and showing it's induced by RA, defining the function of such a factor in the process of P/D specification and execution of patterning seems a better goal in the modern age, and it is achievable. Overall, this report presents preliminary findings, most of which need deeper investigation.

We agree with the reviewer that the 'definition of a mediator of proximal identity', which may have been suggested by the 'set of principles' mentioned in our former introduction, is strong and simply reflects an approach to the difficult problem of identifying candidates for PD determination. This is now reflected in the new version.

Of note, we are not claiming that Tig1 is a “master regulator”; indeed we think it is a downstream effector of RA. But its over-activation is sufficient, independently of RA, to proximalise the transcriptome of distal connective tissue cells on a global/broad scale.

SPECIFIC CONCERNS:

- The identity of Tig1 seems to solely rely on the sequence alignment between the putative axolotl Tig1 and the human Tig1 versus other proteins. Given the similarities of Tig1 with CD38, the authors should strengthen these claims by performing a phylogenetic tree analysis of Tig1 and CD38 genes to verify the proper annotation. Relatedly, it appears that there are indications from the literature (Bryant et al., 2017) that CD38 itself may have a role in P/D axis in regenerating axolotl limbs, and so the authors should include this in their discussion.

Although TIG1 shows similarity at the structural level with CD38, there is a clear separation between the two regarding their sequence, as seen in the phylogenetic tree analysis suggested by the Reviewer and now incorporated in **new Suppl Fig 3**.

CD38 has been identified in the RNA-seq Bryant et al 2017 dataset as a gene enriched in the proximal (upper arm) vs. distal (lower arm) whole blastema tissue. CD38 is retinoic acid-responsive molecule in immune cells, but whether this stands true for axolotl hematopoietic or limb connective tissue is unknown. Since in the mentioned analysis there was no enrichment for connective tissue performed, it is possible that the CD38 expression reflects immune cell populations that infiltrate the blastema during the early stages of regeneration. Given that no further studies were conducted, it remains to be determined whether CD38 partakes any role in conferring proximal identity to cells in the axolotl.

- In other previously-published axolotl gene expression datasets (Campbell et al., Knapp et al., Stewart et al., Voss et al., Bryant et al., Gerber et al., Leigh et al., etc.) what does Tig1 transcript expression look like in those studies?

We identified TIG1 in Knapp et al 2013, Gerber et al 2018 and the more recently published Lin et al 2021 datasets.

In the Knapp et al 2013 dataset, TIG1 transcription increases from day 1 dpa until 3dpa when it reaches a plateau, initiating a gradual decrease from 7dpa on (now represented in **new Fig 2d**). Expression goes back to ‘baseline’ intact limb levels at around 12 dpa and remains low at 22 dpa. TIG1 (termed by its alternative name RARRES1) transcript abundance was analysed in 3, 5, 8, 11 and 18 dpa by Gerber et al 2018 (in Fig3-j in the published manuscript), where from an early upregulation at 3 dpa, it follows a decrease from 8dpa on, which is associated to a reduction in the number of TIG1⁺ cells as well as transcript expression level. Our own analysis of this dataset, performed at 11 dpa, is now included in the **new Suppl Fig 1** and shows that TIG1 is associated with the proximal end of the modelled PD axis, based on the expression profile of the HoxA13⁺ and Meis2⁺ markers.

In the revised version of the manuscript, we include a new bioinformatic analysis of connective tissue-enriched 11 dpa blastema from the recently reported dataset in Lin et al 2018 (see also methods section), which now constitutes the **new Fig 1**. The principal component 1 in our analysis strongly differentiates cell populations that express proximal markers Meis1 and 2 and PROD1, from the ones expressing distal markers HoxA13 and HoxD13. Our results strongly associate TIG1-enriched expression to ‘proximal populations’.

Is there any more information about the other tissues that may express this gene other than fibroblasts?

Through TIG1 in situ hybridization combined with PRRX1⁺ immunostaining of 5- and 7-day blastema samples including stump, we confirmed that TIG1 is majorly expressed in connective tissue (70% of CT cells). This data has been added as **new Suppl figure 9**.

Any indication from these other studies that Tig1 has a graded expression?

Unfortunately, datasets discriminating different PD axis levels from intact tissue are not published in these studies.

Can they find the gene in some of the studies that have centered explicitly on the effects of RA (Voss et al., Nguyen et al.)?

We could not identify TIG1 in these datasets for RA-treated samples, but we note that these studies analysed whole blastema tissues, therefore it is possible that TIG1 expression might have been diluted out and subsequently missed.

- Figure 1a: Can the authors comment on why some of the cells might be highly expressing both Tig1 and Hoxa13 transcripts? The paper's model proposes their expression should be anti-correlated, but there appear to be several example cells on these graphs that show the opposite relationship. Is it possible that the principal component used to order the cells is not the best choice?

This is a valid concern. In the **new Suppl data 2** we provide evidence suggesting that this is not the case, as it is only perceived at cutoffs of extremely low stringency. Further, we performed a Fisher's exact test, which ruled out co-expression between Tig1 and Hoxa13 transcripts even at >1TPM, the less stringent cutoff (incorporated in the corresponding figure legend)

We also provide the data corresponding to the PCA analysis performed in **new Suppl data 1**, which supports our choice of PC3 in 11dpa as a model of PD axis. Further, we note that 11dpa is the optimal timepoint for this analysis because a relatively high fraction of positive cells were found for both, Tig1 and Hoxa13. At earlier timepoints Hox13 is rarely expressed while in later timepoints Tig1 is rarely expressed.

- Figure 1c: The statistics performed are not appropriate. Here a one-way ANOVA with post-hoc tests should be used. The expression appears to be decreasing but the Y-axis scale is misleading, this should follow what was used in Figure 1e. These data are used to validate the graded expression of the gene and they are weak.

We agree that the suggested statistical analysis is a more appropriate choice and now provide the data as suggested (see **new Fig 2**). We also use a bee-swarm representation for further clarity.

The TIG1 gradient as measured by RT-PCR is slightly shallow, yet significant, which is reportedly similar in PROD1 and Meis using RT-PCR. We note that this analysis was performed using whole tissue and not connective tissue only, which might have masked a clearer and more pronounced gradient.

In addition, we note that a TIG1 transcripts are also enriched in proximal elements of the *newt* (*N. viridescens*, **new Suppl Fig 2**) limb, which further strengthens the notion of Tig1 transcripts are enriched in proximal segments.

In situ hybridization should be used in addition to what is presented.

Unfortunately, we could not detect TIG1 transcripts using ISH in mature limb sections. In contrast, 1 to 7 dpa blastemas and stump tissues show strong TIG1 expression while a gradual decrease is observed afterwards, until being sparse or undetectable at 10-14dpa (**new Fig 2e**). Microarray data from Knapp et al 2017 support our observations and show that the levels in the intact limb are similar to those found in a 12 dpa blastema and a 22 dpa regenerate, the later timepoint roughly corresponding to the limit of detection of our own in situ. Therefore, it is possible that ISH technique is not sensitive enough to detect the low basal levels of TIG1 expression in mature limb.

It is also unclear as currently presented in Figure 1c whether there was first a purification or enrichment for connective tissues or cells before generating the cDNA for the three sample sites. If this were performed, then perhaps the authors would see a more robust and convincing proximal enrichment of the transcript. The published prrx1:GFP transgenic axolotls could be used to purify those fibroblasts and then used to generate cDNA from various P/D points along the limb axis. Furthermore, without any experiments to determine if other cells outside of connective tissue express Tig1, readers have no idea if the weak signal reflects the fact that this transcript is expressed in other cell types (and doing something else there?) and the contribution to total expression the authors care about is just swamped out. Purification followed by qPCR as well as a quantitative in situ method that includes full-length limb sections as well as magnified sections from several places highlighting connective tissues along P/D axis should be performed.

Our RT-PCR analysis of TIG1 expression in the intact limb was done using cDNA from whole limb tissue. We thank for noting that this information is missing, and we now added it to the methods section.

We agree that it is possible that enriching for connective tissue prior to quantification of TIG1 transcripts would provide us even clearer results, however this is not possible for us at present given the timeline for animal licenses to perform this experiment (currently circa 1 year to obtain), which exceed the deadline for this revision. Nevertheless, we have successfully combined TIG1 in situ with PRRX1+ immunostaining, and confirmed that the majority of TIG1+ cells overlap with the connective tissue marker (70%), please see **new Suppl Fig 9**.

Unfortunately, and as mentioned above, ISH might not be a sensitive enough method to detect TIG1 expression in the intact limb, likely due to its globally low expression level.

- For Figure 1c and 1e, individual data points should be plotted rather than reporting the data as bar or line graphs without this information available to the reader.

We agree with the Reviewer and present the data in the requested format (**new Fig 2**)

- Figure 1f: The microarray data suggests the peak of Tig1 transcript expression is likely to be around 3 days post-amputation. Why, then, in Supplementary Figure 5 for their own qPCR analysis, did they start the regeneration time course samples at 9 days post-amputation? There's a lot of information missing in Supplementary Figure 5 about the expression of the gene in focus in the 9 days between amputation and first sampling.

This gap has been now filled by our ISH analysis (see comments above), in which samples from 1, 3, 5-, 7-, 10- and 14-days post-amputation, from upper and lower arm were performed.

Notably, our ISH mirror the data from the microarray dataset from Knapp et al 2013 during these timepoints. Further, the downregulation at later stages is consistent with our RT-PCR analysis at later stages.

- Overall, Figure 1 is so much less refined than is needed if there is to be a robust argument made that this factor is a strong candidate for conferring positional information regarding P/D axis in connective tissue cells and that it is responsive (i.e., transcription goes up) to retinoic acid treatment.

We concur with the assessment and made efforts to improve the analysis and visualisation of the data. Figure 1 has now been split into **new Suppl Fig 1 and new Fig 2**. The new Figure 1 consists in an in-depth analysis of the dataset published with Lin et al 2021, in which dataset, in addition to TIG1, we also identified and analysed PROD1. In summary, the analysis corroborates the previously suggested correlation of cells highly enriched in TIG1 to clusters positive for the proximal markers Meis1, 2 and PROD1, and therefore associated with proximal identity. Importantly, the new Figure 2 includes a panel of TIG1 ISH regeneration time course, and individual data points were added to the graphs as requested by the Reviewer.

- Figure 2 and discussion of experiments therein: the effects observed following expression of Tig1 in blastemas using baculovirus do not necessarily support a role for Tig1 in patterning the P/D axis of the limb. Skeletal defects are presented, but there is no convincing argument these are not caused by another mechanism. The simplest prediction would have been that overexpression of a proximalizing factor, which Tig1 is proposed to be, would lead to lengthening of the proximal elements or perhaps reduction of more distal elements at the expense of proximal. However, they observe malformations that cannot simply be explained by a proximalization (such as "malformations of carpals and metacarpals" which are binned separately from shortening or truncations, yet no example photos are shown). The graph gives the reader little solace that the effects are specific to P/D since they have not broken the information down any more than "limb length."

This data would be stronger if they had simply measured all the humerus elements, all the radius/ulna elements, all the carpal elements, and all the digits from all the specimens and reported the data more granularly. Again, the individual data points should be presented rather than bar graphs, and so there is more info than the reader gets to see. Additionally, there is zero consideration given here to whether infection events were uniform across the samples. Disentangling this information and presenting it in a more holistic manner would allow the reader to decide if more extreme phenotypes are attributable to "better" infections, and the authors may even stand to improve the believability of their assertions if they were to simply re-analyze the data they've already collected or do the experiments again in a way that this information can indeed be reported.

We thank the reviewer for the suggestions. We have extended this study to conduct a more in-depth examination, and modified the text to incorporate the suggestions and include the more recent analysis.

In **new Fig 3-b** of the revised version we add additional images of bone/cartilage staining of TIG1-treated limb side by side with the respective contralateral control-treated limb. They exemplify the morphological alterations reported in the study, which occurred with particular incidence in more distal elements. These include digit fusion, digit truncation, carpal and metacarpal malformations; occasionally radius and ulna distal epiphysis were elongated or the bones were fused (updated text in P.5 Line 5 and Fig3-b).

We also break down the information beyond 'limb length', by performing limb segment ratio quantifications by measurement of the skeletal elements (see updated methods section). We confirmed that the whole limb length is significantly different, with TIG1-treated samples measuring smaller than the respective contralateral controls. We also observe a trend towards size reduction in the lower arm and a significant downsizing of the hand. Conversely, the upper arm length is comparable to the control samples. We plotted the data as a kernel distribution (including the individual samples' data points), which allows us to easily visualize the shifts in size of each element, now on **new Fig 3d**.

We note that Prod1 overexpression in axolotl blastemas (Echeverri and Tanaka 2005) results in regenerates displaying resembling phenotypes, such as missing and truncated digits, malformed metacarpals and shortening of skeletal elements. Similar to our observations, elongation of the upper arm, as seen in RA-treated animals, is not reported. This particular phenotype might be due to a dosage effect, since the efficiency of electroporation or viral delivery of constructs in the blastema is unlikely to compare to RA treatment delivered to the whole tissue. It is also possible that the phenotypes of overexpressing TIG1 and PROD1 only account for a part of the effect of RA acid in target tissues, which is known to be rather broad.

The reduced size of the limb and the skeletal patterning defects found in the distal areas of the regenerate points to a perturbation in PD patterning, and is therefore, together with the remaining data presented in the manuscript, suggestive of a role of TIG1 in the establishment of the PD axis during regeneration beyond a general impact on cell cycle regulation. We have now made textual changes to clarify this (see corresponding section in P.5)

- The authors generated a plasmid encoding a mutant Tig1 to use as a loss-of-function control, but it appears that it does not function that way. The use of the mutant does not help further the claims made in the paper. For instance, in the displacement assay presented in Figure 3c, the results show that the wild-type Tig1 has a very small effect to the proximal displacement of blastema cells. The mutant Tig1 has a bigger effect which cannot be explained. Another way to look at this is to simply say that the mutant Tig1 (P155A) has a neomorphic activity with respect to normal Tig1 function and that it could be doing something entirely unrelated to what the wild-type protein normally does. How do we even know this is a loss-of-function mutation rather than a neomorphic mutation? Are they hoping it acts as a dominant-negative? Where's the hard data to support that notion, if so?

We apologise for the insufficient description in the main text and clarify that the TIG1^{P155A} mutant is not a 'loss-of-function' form of TIG1. TIG1^{P155A} is designed to disrupt the only known conserved domain of TIG1, the Latexin-like domain. Latexin is a carboxypeptidase inhibitor with tumour suppression function; the shared conserved domain with TIG1 is known to locate in a protruding region of Latexin, and has been proposed to promote homo- and heterotypical interactions between molecules, but the function is not yet clear.

We show that TIG1^{P155A} shows a similar anti-proliferative effect to that of TIG1^{wt} (**Suppl Fig. 6**), indicating that the latexin domain is dispensable for Tig1's anti-proliferation function. Further, we observe that it slightly enhances proximalisation ability, as evaluated by the displacement assay (new Fig 4), both at frequency and the distance of translocation (**new Fig4b-d, and Suppl Fig 12 - p-values graphs**). In our RNAseq experiment we didn't observe significant overall differences pertaining PD-axis related pathways between TIG1^{wt} and TIG1^{P155A}, suggesting that changes at the gene expression level do not explain the phenotype observed. However we note that in the context of proximalisation, this information may be of interest in future studies exploring the molecular mechanisms of Tig1 function. Further, it might be of interest for researchers currently investigating TIG1 domains and their functional relevance.

We added additional considerations regarding TIG1^{P155A} to the main text, for clarification.

Another issue here is that the reader is left with approximating whether the differences in percentages among the different samples are statistically significant only by visual observation let alone assessing whether the difference between the percentages on the lower arm for RFP and TIG1^{wt} have a biological meaning and support the authors claims that TIG1 induces distal-to-proximal displacement of blastema cells.

The use of the TIG1 mutant, in this case, may be to distract the reader from the main issue and that is that the function and the role of wild-type TIG1 and its importance in normal P/D patterning, whereas the “loss-of-function” mutant should show results similar to that of RFP and not the opposite. Overall, looking at the fluorescent information at Figure 3d, Supplementary Figures 6 & 7, or the summarized data in Figure 3b, the effect of TIG1 appears minimal.

The authors should consider strengthening these data with some statistical analysis otherwise it is hard to extract any actual biological meaning.

TIG1^{P155A} is NOT a ‘loss-of-function’ mutant, but a TIG1 version that HAS proximalising activity, and this is supported by all datasets in this manuscript. If anything, as both versions promote proximalisation, it reassures the reader that TIG1 is indeed important for PD patterning. We do understand that this comment arose from an initial misunderstanding, which we hope we have now fully clarified.

Further, in order to provide a measure of significance regarding proximalisation of cells between conditions, we now plotted the p-values, calculated from the output of Meandros software analysis. These plots are now available (**new Suppl Fig 12**) and demonstrate that TIG1^{wt} and TIG1^{P155A} show significantly different fluorescence signal detection in areas more proximal than the ones detected in the control, demonstrating that either TIG1^{wt} or TIG1^{P155A}-transformed cells significantly change their position to more proximal locations along the PD axis of the limb. No significant difference between TIG1^{wt} or TIG1^{P155A} is detected (**new Suppl Fig 12**)

- The data in Figure 4 should be parsed with more granularity. Choosing to bin the results as either “engulfment” or “no engulfment” is misleading (because we can see intermingling at the cellular level even in the photos presented for what they claim is the most extreme case of non-engulfment, application of the TIG1 antibody). The authors should assay this result with more granularity. What percentage of RFP+ cells end up in the distally-derived tissue in each experimental scenario? Also, what is going on with the vastly larger tissue sizes in the examples treated with the antibody to TIG1? Were these blastemas bigger from the beginning, or are they larger by virtue of more proliferation following explant? Or are these images not representative of the whole cohort examined? As presented, the first thing that could be concluded, if this antibody truly blocks TIG1 function, is that TIG1 might be a candidate tumour suppressor.

We thank again for the comments. We want to clarify that the panel in the **old Fig 4 (now Fig 5b)** shows representative images of blastema pairs for each different condition, which belong to the same assay (same replicate) and therefore were subjected to the exact same incubation conditions. Also, the blastemas were collected from size-matched animals from the same clutch and all had comparable sizes in the beginning of the experiment. The dissimilar sizes observed between conditions at the later timepoints are results of biological processes acting between the pairs as a function of their treatment. Specifically, we noted that after 4-5 days in culture, the control antibody-treated pairs consistently undergo apparent condensation resulting in a sphere-like structure. In contrast, anti-TIG1 antibody-treated blastemas generally remain larger.

Taking into account the Reviewer’s suggestions, we repeated the control and anti-TIG1 antibody incubation experiments using blastemas from GFP transgenic animals and blastemas from white wild type animals (siblings from the same batch, through *caggs:GFP* x wildtype cross), a combination that allowed to have a more clear and consistent labelling of the proximal blastemas. At 5 days post juxtaposition, we already observed significant engulfment in most of the control-treated pairs, while the original boundary between proximal and distal blastemas remained well defined. This was confirmed and is clearly visualized in the sample’s cryosectioned images, now included as **new Fig 5d**.

Quantification of the overlap area created by the engulfment as a percentage of the total pair area, revealed a 19% proximal overlap in the control versus only 2% in the anti-TIG1 antibody treated pairs condition, highlighting a very significant difference between the treatments. We added the plot to **new Fig. 5e**, and updated the main text accordingly.

Furthermore, we combined the incubations with EdU incorporation (one pulse, 16h incubation) to test the hypothesis that the larger size of the anti-TIG1 antibody-treated blastema pairs observed after 5 days in culture was due to a rescue of proliferation caused by antibody-mediated 'neutralisation' of TIG1-induced anti-proliferative effects. Our results in the graph below show that, instead, fewer cells in the anti-TIG1 antibody treated condition enter the S-phase, therefore not explaining the difference in size. To check for alterations at the level of overall compaction between cells, we quantified the nuclei density, but we found no statistical difference between conditions.

Figure 5 – EdU incorporation of juxtaposed proximal/distal blastemas after 5 days in culture, a time point where engulfment is readily visible in the control condition.

Another issue is specificity of the reagent. Is aRFP the correct control for the engulfment assay? An antibody that binds to a cell surface protein and is considered not be functioning during this process should be used as control instead in order to test whether just having antibodies on cells affect their migration.

We think that an antibody that would bind a surface molecule could impact on our results, either by creating an additional phenotype or by simply physically impeding the establishment of required cell-cell contacts that underlie the mechanism of engulfment behaviour. Therefore, we expect an anti-RFP antibody, to be a more adequate control.

- The antibodies used have not been adequately characterized for readers to believe they are specific. Spec sheets from the company commissioned to generate the antibodies were provided, but these are irrelevant to the application of these antibodies. Evidence the antibodies are specific on western blot from protein extracted from axolotl tissues, on immunohistochemistry of axolotl tissues, and on the behavior of cells in vitro is lacking. The authors have made no attempt to perform peptide-blocking experiments that could help address these concerns; for example, any attempt to out-compete the supposed effect of the antibody in the engulfment assay with the peptide that served as the antigen.

We conducted western blot analysis of the antibodies, which demonstrate their specificity. Please see **new Suppl Fig 15a**. Further, as suggested by the Reviewer, we performed in vitro analysis of the antibody. By performing a wound healing assay/scratch assay, we observe that TIG1 transfected cells display less efficient closure of the gap area generated by a partition, which is partially rescued using anti-TIG1 antibodies. This result now in **new Suppl Fig 15b-d**. This indicates that TIG1 induces changes at the level of cell migration or cell-cell interaction, consistent with reports in mammalian systems (Wang et al, 2013), and is highly relevant to the interpretation of the effects of Tig1 disruption in our engulfment assay (see additions in the main text).

- Again, in figure 5, there needs to be a better justification for believing the P155A mutation is effectually a loss-of-function or dominant-negative allele of Tig1. See previous comment about this. There are a variety of other interpretations for the data they present besides an overt role in orchestrating proximal-distal patterning in regeneration. The HoxA13:mCherry transgenic axolotl line is a nice addition to the paper and the field. Are the transgenics used non-mosaic? It was hard to tell from the methods if this is the case or not. Certainly, F0/mosaic animals would not be appropriate for this kind of experiment because we need to believe that all cells have the knock-in allele and that only some are turning the fluorophore on by virtue of them also expressing HoxA13 rather than the more trivial explanation that they were simply not edited at either HoxA13 allele.

The P155A mutation is not a loss-of-function or dominant-negative version, please see comments above.

The HoxA13:mCherry transgenic animals used are F2 generation.

- Tig1 overexpression does not affect meis1 expression as shown from the qPCR data in Figure 6 and had variable expression in the RNAseq dataset (no standard deviation bars are presented in Figure 5). Figure 6 shows no effect on mesi1, and minimal effect on prod1, so the author's claims are not supported. The statistics for Figure 6 are not appropriate. One-way ANOVA with post hoc tests should be used instead if the variances among the samples are equal and equivalent non-parametric tests should be used if they are not (which in many of them seem to be the case based on the standard deviation presented here).

We agree with the Reviewer and have now performed one-way Welch ANOVA tests, followed by Welch t-tests for individual comparisons in all cases where a significant interaction was indicated by the Welch ANOVA. We also annotate the p values for the Welch ANOVA test on top of each group analysed. Please see the **new Fig 7**.

- The authors should mention that the statistics presented for the RNAseq expression data are q-values and not p-values as indicated in the legend (Figure 5).

We presented our analysis with 'adjusted p-values' (means that they are adjusted for the False Discovery Rate) which is the same as q-values. The two terms are synonyms.

- An integral issue with the mCherry+ (HOXA13+) sorting of the cells and RNAseq is that the Tig1 overexpression downregulates the HOXA13 expression which controls the mCherry. In addition, Tig1+ cells will assume more proximal positions in the blastema (according to the authors data and claims), thus the "reprogramming" of their transcriptomic profiles may be not due to the direct effect of the Tig1 overexpression but rather the effects of other PD-related molecules including retinoic acid.

We already observe changes at the transcriptional level at 3 days, which is too early for the cells to undergo translocation to proximal sites.

Concerning the first consideration, we think that if a drastic decrease in mCherry was to happen, it would indicate that our data represents an underestimate of the TIG1-induced effects. But as we note in the main text, mCherry is a very stable fluorophore that in our hands, is detectable over 20 days after RNA expression is no longer detected.

Reviewers' Comments:

Reviewer #1:

Remarks to the Author:

I appreciate the effort put in by the authors to improve the manuscript, which is much improved and suitable in my opinion for publication.

Reviewer #2:

Remarks to the Author:

The authors responded in detail to all questions and concerns from the reviewers. Although the authors were not able to perform experiments to answer all concerns, they provided sufficient explanation and additional experiments to the study to warrant publication of the manuscript.

Reviewer #3:

Remarks to the Author:

Some aspects of the paper have been clarified and/or improved. However, there remain some outstanding issues that should be fixed before publication.

REMAINING SPECIFIC CONCERNS:

- Figure 1c: In situ hybridization should be used in addition to what is presented.

All three reviewers asked for this. They have now performed an in situ series starting at 1dpa and going through 14 dpa. The in situ is incredibly weak and unconvincing. This means the graded expression of Tig1 transcripts as a result hinges totally on the qPCR (which has its own issues, see below) and the bioinformatics. The bioinformatics is somewhat consistent, but then again one of the analyses shows Prod1 expression to be highest in the middle of the P/D axis. So we really do need convincing experimental data, in the form of the in situ, to be convinced that Tig1 is expressed in a gradient along P/D axis and even that the blastema expresses the transcript. Perhaps a different probe? And/or maybe RNAscope or FISH?

- Figure 1f: The microarray data suggests the peak of Tig1 transcript expression is likely to be around 3 days post-amputation. Why, then, in Supplementary Figure 5 for their own qPCR analysis, did they start the regeneration time course samples at 9 days post-amputation? There's a lot of information missing in Supplementary Figure 5 about the expression of the gene in focus in the 9 days between amputation and first sampling.

The authors now claim to have filled this gap, but they want to say the gap is filled with the in situ hybridization. The in situ is not convincing (separate issue), but the more appropriate experiment to do anyway is indeed the qPCR from these other time points. That makes a lot more sense and is actually quantitative, whereas even if the in situ were convincing in its results (as in, robust staining and in the places the model predicts), it does not provide the direct comparison asked for.

- The data in Figure 4 . . . Is aRFP the correct control for the engulfment assay? An antibody that binds to a cell surface protein and is considered not be functioning during this process should be used as control instead in order to test whether just having antibodies on cells affect their migration.

The authors decided not to perform an experiment to address the specificity of the anti-Tig1 result in the engulfment assay. Their explanation seems counter-intuitive; if they are concerned that any antibody to any protein expressed on the cell surface will lead to inhibition of engulfment, then why do they also think their result is specific/meaningful? The response from the authors makes the assay seem irrelevant for having meaning with respect to setting P/D axis in vivo. If there is true biological meaning for Tig1 in the process of determining P/D axis in vivo, and this engulfment assay reflects this, then not all cell-surface proteins will be expected to directly control engulfment. The anti-RFP antibody does not bind a cell-surface protein because RFP is presumably restricted to the intracellular component of these cells. Performing the assay with an antibody to a well-established cell-surface protein expressed by these cells makes more sense and is a reasonable control to perform that will address specificity of the core finding of the paper.

REVIEWER COMMENTS

Reviewer #1 (Remarks to the Author):

I appreciate the effort put in by the authors to improve the manuscript, which is much improved and suitable in my opinion for publication.

We thank the Reviewer for appreciating our efforts towards improving our manuscript.

Reviewer #2 (Remarks to the Author):

The authors responded in detail to all questions and concerns from the reviewers. Although the authors were not able to perform experiments to answer all concerns, they provided sufficient explanation and additional experiments to the study to warrant publication of the manuscript.

We thank the Reviewer for his/her assessment and for considering the manuscript publication-ready.

Reviewer #3 (Remarks to the Author):

Some aspects of the paper have been clarified and/or improved. However, there remain some outstanding issues that should be fixed before publication.

REMAINING SPECIFIC CONCERNS:

- Figure 1c: In situ hybridization should be used in addition to what is presented.

All three reviewers asked for this. They have now performed an in situ series starting at 1 dpa and going through 14 dpa. The in situ is incredibly weak and unconvincing. This means the graded expression of Tig1 transcripts as a result hinges totally on the qPCR (which has its own issues, see below) and the bioinformatics. The bioinformatics is somewhat consistent, but then again one of the analyses shows Prod1 expression to be highest in the middle of the P/D axis. So we really do need convincing experimental data, in the form of the in situ, to be convinced that Tig1 is expressed in a gradient along P/D axis and even that the blastema expresses the transcript. Perhaps a different probe? And/or maybe RNAscope or FISH?

We agree with the Reviewer that the images s/he received (Fig 2) were of poor quality and suggestive of weak signal. While the images we submitted were at high-resolution, the later was severely decreased upon conversion by the submission system, something which escaped our notice upon submission checks and for which we apologize. We now present the .ai files, in which clear Tig1 signal can be appreciated at single-cell resolution.

Further to this, we performed an additional ISH experiment on new samples and present representative images of key timepoints at high resolution and in a large-panel format (new Supplementary Fig. 9- upper arm- and 11 -lower arm-).

Together, these demonstrate that within the blastema Tig1 expression is strongest at the proximal end, decreasing towards distal locations.

- Figure 1f: The microarray data suggests the peak of Tig1 transcript expression is likely to be around 3 days post-amputation. Why, then, in Supplementary Figure 5 for their own qPCR analysis, did they start the regeneration time course samples at 9 days post-amputation? There's a lot of information missing in Supplementary Figure 5 about the expression of the gene in focus in the 9 days between amputation and first sampling.

The authors now claim to have filled this gap, but they want to say the gap is filled with the *in situ* hybridization. The *in situ* is not convincing (separate issue), but the more appropriate experiment to do anyway is indeed the qPCR from these other time points. That makes a lot more sense and is actually quantitative, whereas even if the *in situ* were convincing in its results (as in, robust staining and in the places the model predicts), it does not provide the direct comparison asked for.

As clarified above, our original *in situ* hybridization data suffered from technical issues upon pdf conversion by the submission system that resulted in decreased image quality, something which we have rectified and expanded with additional examples in this new version (see previous point).

In addition, we have considered the reviewer's request and we have now performed a qRT-PCR analysis of the timepoints that correspond to the early stages of blastema formation. In our initial analysis, our aim was to specifically profile the blastema, which could only be confidently isolated from 9 dpa due to the low amount of protruding mesenchymal blastema tissue.

According to the reviewers request and in order to address the transcriptional changes in the blastema cells at early timepoints, we now collected blastema including a small portion of the proximal adjacent tissue that corresponds to the blastemal transition area created upon amputation. Our results, plotted in the graph below, demonstrate a significant upregulation of TIG1 expression at early stages of blastemal formation, and validates our microarray analysis.

From new Supplementary Fig 7. TIG1 expression levels in the axolotl upper arm blastema including 500 μ m of tissue proximal to the amputation plane at the indicated times after amputation. Expression is normalised to Rlp4 and visualised as fold change relative to the intact limb (mature). The chart shows the means, standard deviations and adjusted p-values corresponding to Dunnet's multiple comparison test against mature limb following 2-way ANOVA (n=3-4). The individual points represent the means of the technical replicates.

- The data in Figure 4 . . . Is aRFP the correct control for the engulfment assay? An antibody that binds to a cell surface protein and is considered not be functioning during this process should be used as control instead in order to test whether just having antibodies on cells affect their migration.

The authors decided not to perform an experiment to address the specificity of the anti-TIG1 result in the engulfment assay. Their explanation seems counter-intuitive; if they are

concerned that any antibody to any protein expressed on the cell surface will lead to inhibition of engulfment, then why do they also think their result is specific/meaningful? The response from the authors makes the assay seem irrelevant for having meaning with respect to setting P/D axis in vivo. If there is true biological meaning for Tig1 in the process of determining P/D axis in vivo, and this engulfment assay reflects this, then not all cell-surface proteins will be expected to directly control engulfment. The anti-RFP antibody does not bind a cell-surface protein because RFP is presumably restricted to the intracellular component of these cells. Performing the assay with an antibody to a well-established cell-surface protein expressed by these cells makes more sense and is a reasonable control to perform that will address specificity of the core finding of the paper.

Upon the Reviewer's request, we have now performed the engulfment assay using an antibody against TMEFF1/tomoregulin-1 (previously used in Da Silva et al), a well-established cell surface protein not involved in engulfment. As expected, anti-tomoregulin treatment does not affect the engulfment activity, in contrast to anti-Tig1 treatment (old Fig 4 – new Fig 5c). Further to this, we have performed an additional control with highly diluted anti-Tig1 antibody (1:200, also a commonly used control – Da Silva et al), which also does not impair engulfment (new Fig 5c). Together, these controls demonstrate the specificity of the anti-Tig1 antibody treatment in disrupting the engulfment activity of proximal blastemas.

Blastema pair	Engulfment	
	Frequency	%
PD	12/17	70.5
PP	1/7	14.2
DD	0/3	0.0
PD + α RFP	6/8	75.0
PD + α TIG1	0/10	0.0
PD + α TIG1(1:200)	4/5	80.0
PD + α PROD1	3/10	30.0
PD + α TMEFF1	6/9	66.7

Form new Fig 5c. Quantification of engulfment for the indicated pairs (pooled data from 3 independent experiments). For α -Prod1, α -Tig1 and α -TMEFF1, [antibody]=20 μ g/ml.

Representative images of juxtaposed proximal (fluorescent) and distal (white) blastemas at the indicated times post juxtaposition, and after the indicated treatments. Note engulfment following treatment with anti-TIG1 antibodies diluted 1:200, as well as with antibodies raised against the extracellular EGF-like domain of TMEFF1. [TMEFF1]=20 μ g/ml. Scale bar: 500 μ m.

We thank the Reviewer for his/her comments, which contributed to improving our work.

Reviewers' Comments:

Reviewer #3:

Remarks to the Author:

The authors have improved the paper by adding qPCR for some of the claims. They have also attempted to better control for non-specific antibody effects in the engulfment assay, although performing the additional experiments alongside fresh versions of the original ones would have been preferable.